# POSTERIOR LABEL SMOOTHING FOR NODE CLASSIFICATION

## ABSTRACT

Soft labels can improve the generalization of a neural network classifier in many domains, such as image classification. Despite its success, the current literature has overlooked the efficiency of label smoothing in node classification with graph-structured data. In this work, we propose a simple yet effective label smoothing for the transductive node classification task. We design the soft label to encapsulate the local context of the target node through the distribution of the neighborhood label. We apply the smoothing method for seven baseline models to show its effectiveness. The label smoothing methods improve the classification accuracy in 10 node classification datasets in most cases. In the following analysis, we find that incorporating global label statistics in posterior computation is the key to the success of label smoothing. Further investigation reveals that the soft labels mitigate overfitting during training, leading to better generalization performance. Our code is available at https://anonymous.4open.science/r/PosteL.

## 1 INTRODUCTION

Adding a uniform noise to the ground truth labels has shown remarkable success in training neural networks for various classification tasks, including image classification and natural language processing (Szegedy et al., 2016; Vaswani et al., 2017; Müller et al., 2019; Zhang et al., 2021a). Despite its simplicity, label smoothing acts as a regularizer for the output distribution and improves generalization performance (Pereyra et al., 2017). More sophisticated soft labeling approaches have been proposed based on the theoretical analysis of label smoothing (Li et al., 2020; Lienen & Hüllermeier, 2021).

In the graph domain, soft labels have been employed to improve the performance of node classification tasks. Based on the homophilic assumption, where the nodes with the same label are likely to be connected, previous studies often employed neighborhood labels to soften the ground truth labels (Wang et al., 2021; Zhou et al., 2023). Despite the success of these approaches, their performance on heterophilic graphs, where nodes tend to connect with others that are dissimilar or belong to different classes, still remains questionable (Zhu et al., 2021; Luan et al., 2022; Chanpuriya & Musco, 2022).

In this work, we propose a *simple yet effective* smoothing method for transductive node classification that can be used for both homophilic and heterophilic graphs. Inspired by the previous work suggesting predicting the local context of a node (Hu et al., 2019; Rong et al., 2020), such as subgraph prediction, helps to learn better representations, we propose a smoothing method that can potentially reflect the local context of the target node. To encode the neighborhood information into the node label, we propose to relabel the node with a posterior distribution of the label given neighborhood labels.

Under the assumption that the neighborhood labels are conditionally independent given the label of the node to be relabeled, we factorize the likelihood into the product of conditional distributions between two adjacent nodes. To compute the posterior, we estimate the conditionals and prior from a graph's global label statistics, making the posterior incorporate the local structure and global label distributions. Since the posterior obtained in this way does not preserve the ground truth label, we finally interpolate the posterior with the ground truth label, resulting in a soft label.

The posterior, however, may pose high variance when there are few numbers of neighborhood nodes. To mitigate the issue with the sparse labels, we further propose iterative pseudo labeling to re-estimate the likelihood and prior based on the pseudo labels. Specifically, we use the pseudo labels of validation and test sets to update the likelihood and prior, along with the ground truth labels of the training set.

We apply our smoothing method to seven different baseline neural network models, including MLP and variants of graph neural networks, and test its performance on ten benchmarks, including homophilic and heterophilic graphs. Our empirical study finds that the soft label with iterative pseudo labeling improves the accuracy in 76 out of 80 cases despite its simplicity. We analyze the cases where the soft label decreases the accuracy and reveals characteristics of label distributions with which the soft labeling may not work. Further analysis shows that using local neighborhood structure and global label statistics is the key to its success. Through the loss curve analysis, we find that the soft label prevents overfitting, leading to a better generalization performance in classification.

## 2 RELATED WORK

In this section, we introduce previous studies related to our method. We begin by discussing various node classification methods, followed by an exploration of the application of soft labels in model training.

### 2.1 NODE CLASSIFICATION

Graph structures are utilized in various ways for node classification tasks. Some studies propose model frameworks based on the assumption of specific graph structures. For example, GCN (Kipf & Welling, 2016), GraphSAGE (Hamilton et al., 2017), and GAT (Veličković et al., 2017) aggregate neighbor node representations based on the homophilic assumption. To address the class-imbalance problem, GraphSMOTE (Zhao et al., 2021), ImGAGN (Qu et al., 2021), and GraphENS (Park et al., 2022) are proposed for homophilic graphs. $H_2$GCN (Zhu et al., 2020) and U-GCN (Jin et al., 2021) aggregate representations of multi-hop neighbor nodes to improve performance on heterophilic graphs. Other studies concentrate on learning graph structure. GPR-GNN (Chien et al., 2020) and CPGNN (Zhu et al., 2021) learn graph structures to determine which nodes to aggregate adaptively. Besides, research such as ChebNet (Defferrard et al., 2016), APPNP (Gasteiger et al., 2018), and BernNet (He et al., 2021) focus on learning appropriate filters from the graph signals.

### 2.2 CLASSIFICATION WITH SOFT LABELS

Hinton et al. (2015) demonstrate that a small student model trained using soft labels generated by the predictions of a large teacher model shows better performance than a model trained using one-hot labels. This approach, known as knowledge distillation (KD), is recognized as effective for compression or performance improvement (Liu et al., 2019; Jiao et al., 2020; Tang & Wang, 2018).

On the other hand, simpler alternatives to generate soft labels are considered. The label smoothing (LS) (Szegedy et al., 2016) generates soft labels by adding uniform noise to the labels. The benefits of LS have been widely explored. Müller et al. (2019) show that LS improves model calibration. Lukasik et al. (2020) establish a connection between LS and label-correction techniques, revealing LS can address label noise.

While label smoothing has been widely adopted in computer vision (Zhang et al., 2021a; Lukov et al., 2022; Vasudeva et al., 2024) and NLP (Vaswani et al., 2017; Song et al., 2020; Guo et al., 2021), the efficiency of smoothing in the graph domain has been less explored. To the best of our knowledge, there are two papers that propose label smoothing methods for node classification. SALS (Wang et al., 2021) proposes a method for smoothing node labels to make them more similar to the labels of neighboring nodes. Similarly, ALS (Zhou et al., 2023) generates soft node labels by aggregating neighborhood labels and applying adaptive label refinement. Both methods rely on the homophilic assumption that connected nodes should have similar labels, which may negatively impact performance on heterophilic graphs (Zhu et al., 2021; Luan et al., 2022; Chanpuriya & Musco, 2022).

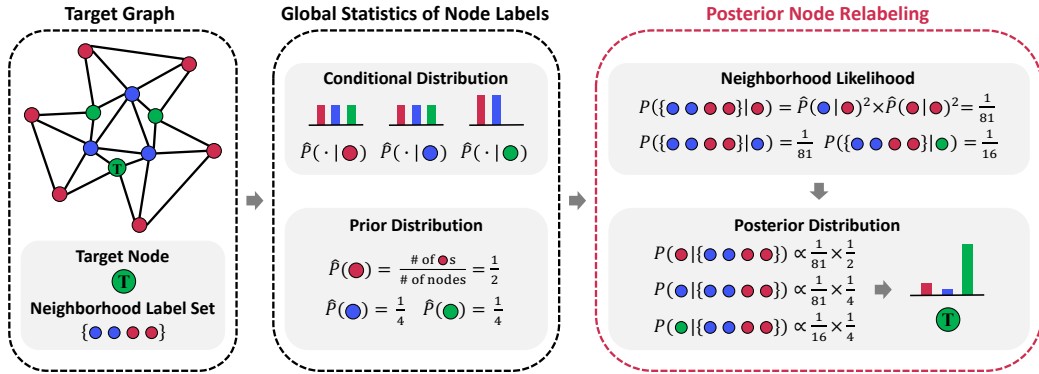

Figure 1: Overall illustration of posterior node relabeling. To relabel the node label, we compute the posterior distribution of the label given neighborhood labels. Note that the node features are not considered in the relabeling process.

Meanwhile, smoothing at the prediction output has been proposed (Zhang et al., 2021b; Xie et al., 2023) to adjust the final prediction based on a graph structure. The motivation of these approaches is significantly different from the label smoothing discussed in this paper.

## 3 METHOD

In this section, we describe our approach for label smoothing for the node classification problem and provide a new training strategy that iteratively refines the soft labels via pseudo labels obtained from the training procedure.

### 3.1 POSTERIOR LABEL SMOOTHING

Consider a transductive node classification with graph $\mathcal{G} = (\mathcal{V}, \mathcal{E}, \boldsymbol{X})$, where $\mathcal{V}$ and $\mathcal{E}$ denotes the set of nodes and edges respectively, and $\boldsymbol{X} \in \mathbb{R}^{|\mathcal{V}| \times d}$ denotes $d$-dimensional node feature matrix. For each node $i$ in a training set, we have a label $y_i \in [K]$, where $K$ is the total number of classes. We use the notation $\boldsymbol{e}_i \in \{0, 1\}^K$ for one-hot encoding of $y_i$, i.e., $e_{ik} = 1$ if $y_i = k$ and $\sum_k e_{ik} = 1$. In a transductive setting, we observe the connectivity between all nodes, including the test nodes, without having true labels of the test nodes.

We propose a simple and effective relabeling method to allocate a new label of a node based on the label distribution of the neighborhood nodes. Specifically, we consider the posterior distribution of node labels given their neighbors. Let $\mathcal{N}(i)$ be a set of neighborhood nodes of node $i$. If we assume the distribution of node labels depends on the graph connectivity, then the posterior probability of node $i$'s label, given its neighborhood labels, is

$$P(Y_i = k | \{Y_j = y_j\}_{j \in \mathcal{N}(i)}) = \frac{P(\{Y_j = y_j\}_{j \in \mathcal{N}(i)} | Y_i = k) P(Y_i = k)}{\sum_{\ell=1}^{K} P(\{Y_j = y_j\}_{j \in \mathcal{N}(i)} | Y_i = \ell) P(Y_i = \ell)} . \tag{1}$$

The likelihood measures the joint probability of the neighborhood labels given the label of node $i$. To obtain the likelihood, we assume that the neighborhood labels are conditionally independent given the label of the node to be relabeled. The likelihood is then approximated by the product of empirical conditional label distribution between adjacent nodes, i.e., $P(\{Y_j = y_j\}_{j \in \mathcal{N}(i)} | Y_i = k) \approx \prod_{j \in \mathcal{N}(i)} P(Y_j = y_j | Y_i = k, (i, j) \in \mathcal{E})$, where $P(Y_j = y_j | Y_i = k, (i, j) \in \mathcal{E})$ is the conditional of between adjacent nodes. The conditional between adjacent nodes $i$ and $j$ with label $n$ and $m$, respectively, is estimated by

$$\hat{P}(Y_j = m | Y_i = n, (i, j) \in \mathcal{E}) := \frac{|\{(u, v) \mid y_v = m, y_u = n, (u, v) \in \mathcal{E}\}|}{|\{(u, v) \mid y_u = n, (u, v) \in \mathcal{E}\}|} . \tag{2}$$

The prior distribution is also estimated from the empirical observations. We use the empirical proportion of label as a prior, i.e., $\hat{P}(Y_i = m) := |\{u \mid y_u = m\}| / |\mathcal{V}|$. We also explore alternative designs for the likelihood and compare their performances in Section 4.2.

Note that, in implementation, all empirical distributions are computed only with the training nodes and their labels. The empirical distribution might be updated after node relabeling through the posterior computation, but we keep it the same throughout the relabeling process.

The posterior distribution can be used as a soft label to train the model, but we add uniform noise $\epsilon$ to the posterior to mitigate the risk of the posterior becoming overly confident if there are few or no neighbors. In addition, since the most probable label from the posterior might be different from the ground truth label, we interpolate the posterior with the ground truth label. To this end, we obtain the soft label $\hat{e}_i$ of node $i$ as

$$\hat{e}_i = \alpha\tilde{e}_i + (1-\alpha)e_i , \qquad (3)$$

where $\tilde{e}_{ik} \propto P(Y_i = k \mid \{Y_j = y_j\}_{j \in \mathcal{N}(i)}) + \beta\epsilon$. $\alpha$ and $\beta$ control the importance of interpolation and uniform noise. By enforcing $\alpha < 1/2$, we can keep the most probable label of soft label the same as the ground truth label, but we find that this condition is not necessary in empirical experiments. We name our method as PosteL (**Poste**rior **L**abel smoothing). The detailed algorithm of PosteL is shown in Algorithm 1. We provide an in-depth analysis of the underlying assumptions of PosteL and an analysis of PosteL's characteristics in heterophilic graphs in Appendix A.

---

**Algorithm 1** PosteL: Posterior label smoothing

---

**Require:** The set of training nodes $\mathcal{V}_{\text{train}} \subset \mathcal{V}$, the number of classes $K$, one-hot encoding of training node labels $\{e_i\}_{i \in \mathcal{V}_{\text{train}}}$, and hyperparameters $\alpha$ and $\beta$.
**Ensure:** The set of soft labels $\{\hat{e}_i\}_{i \in \mathcal{V}_{\text{train}}}$.
   Estimate prior distribution for $m \in [K]$: $\hat{P}(Y_i = m) = \sum_{u \in \mathcal{V}_{\text{train}}} e_{um}/|\mathcal{V}_{\text{train}}|$.
   Define the set of training neighbors for each node $u$: $\mathcal{N}_{\text{train}}(u) = \mathcal{N}(u) \cap \mathcal{V}_{\text{train}}$.
   Estimate the empirical conditional for $n, m \in [K]$:

$$\hat{P}(Y_j = m | Y_i = n, (i,j) \in \mathcal{E}) \propto \sum_{u:u \in \mathcal{V}_{\text{train}}, y_u = n} \sum_{v \in \mathcal{N}_{\text{train}}(u)} e_{vm}.$$

   **for** each $i \in \mathcal{V}_{\text{train}}$ such that $\mathcal{N}_{\text{train}}(i) \neq \emptyset$ **do**
      Approximate likelihood:

$$P(\{Y_j = y_j\}_{j \in \mathcal{N}_{\text{train}}(i)} | Y_i = k) \approx \prod_{j \in \mathcal{N}_{\text{train}}(i)} \hat{P}(Y_j = y_j | Y_i = k, (i,j) \in \mathcal{E}).$$

      Compute posterior distribution: $P(Y_i = k \mid \{Y_j = y_j\}_{j \in \mathcal{N}_{\text{train}}(i)})$ using Equation (1).
      Add uniform noise: $\tilde{e}_{ik} \propto P(Y_i = k \mid \{Y_j = y_j\}_{j \in \mathcal{N}_{\text{train}}(i)}) + \beta\epsilon$.
      Obtain soft label: $\hat{e}_i = \alpha\tilde{e}_i + (1-\alpha)e_i$.
   **end for**

---

## 3.2 ITERATIVE PSEUDO LABELING

Posterior relabeling is a method used to predict the label of a node based on the labels of its neighboring nodes. However, in transductive node classification tasks where train, validation, and test nodes coexist within the same graph, the presence of unlabeled nodes can hinder the accurate prediction of posterior labels. For instance, when a node has no labeled neighbors, the likelihood becomes one, and the posterior only relies on the prior. Moreover, in cases where labeled neighbors are scarce, noisy labels among the neighbors can significantly compromise the posterior distribution. Such challenges are particularly prevalent in sparse graphs. For example, 26.35% of nodes in the Cornell dataset have no neighbors with labels. In such scenarios, the posterior relabeling can be challenging.

To address these limitations, we propose to update the likelihoods and priors through the pseudo labels of validation and test nodes. We first train a graph neural network with the soft labels obtained via Equation (3) and predict the labels of validation and test nodes to obtain the pseudo labels. We choose the most probable label as a pseudo label from the prediction. We then update the likelihood and prior with the pseudo labels of the validation and test nodes while keeping the ground-truth labels of the training nodes. This process re-calibrates the posterior smoothing and soft labels. By repeating training and re-calibration until the best validation loss of the predictor no longer decreases, we can maximize the performance of node classification. We assume that if posterior label smoothing improves classification performance with a better estimation of likelihood and prior, the pseudo labels obtained from the predictor can benefit the posterior estimation as long as there are

not many false pseudo labels. The detailed algorithms for PosteL using pseudo labels, in addition to the training process involving iterative pseudo labeling, are shown in Algorithm 2 and Algorithm 3 in Appendix B. Furthermore, we discuss the distinct behavior of PosteL compared to SALS and ALS in Appendix C.

# 4 EXPERIMENTS

The experimental section is composed of two parts. First, we evaluate the performance of our method for node classification through various datasets and models. Second, we provide a comprehensive analysis of our method, investigating the conditions under which it performs well and the importance of each design choice.

## 4.1 NODE CLASSIFICATION

In this section, we assess the enhancements in node classification performance across a range of datasets and backbone models. Our aim is to validate the consistent efficacy of our method across datasets and backbone models with diverse characteristics.

**Datasets** We assess the performance of our method across 10 node classification datasets. To examine the effect of our method on diverse types of graphs, we conduct experiments on both homophilic and heterophilic graphs. Adjacent nodes in a homophilic graph are likely to have the same label. Adjacent nodes in a heterophilic graph are likely to have different labels. For the homophilic datasets, we use five datasets: the citation graphs Cora, CiteSeer, and PubMed (Sen et al., 2008; Yang et al., 2016), and the Amazon co-purchase graphs Computers and Photo (McAuley et al., 2015). For the heterophilic datasets, we use five datasets: the Wikipedia graphs Chameleon and Squirrel (Rozemberczki et al., 2021), the Actor co-occurrence graph Actor (Tang et al., 2009), and the webpage graphs Texas and Cornell (Pei et al., 2020). Detailed statistics of each dataset are illustrated in Appendix D.

**Experimental setup and baselines** We evaluate the performance of PosteL across various backbone models, ranging from MLP, which ignores underlying structure between nodes, to seven widely used graph neural networks: GCN (Kipf & Welling, 2016), GAT (Veličković et al., 2017), APPNP (Gasteiger et al., 2018), ChebNet (Defferrard et al., 2016), GPR-GNN (Chien et al., 2020), BernNet (He et al., 2021), and OrderedGNN (Song et al., 2023). We follow the experimental setup and backbone implementations of He et al. (2021). Specifically, we use fixed 10 train, validation, and test splits with ratios of 60%/20%/20%, respectively, and measure the accuracy at the lowest validation loss. The model is trained for 1,000 epochs, and we apply early stopping when validation loss does not decrease during the last 200 epochs. We report the mean performance and 95% confidence interval. The detailed experimental setup, including the search spaces of the hyperparameters, is provided in Appendix E.

We compare our method with two domain-agnostic soft labeling methods, including label smoothing (LS) (Szegedy et al., 2016) and knowledge distillation (KD) (Hinton et al., 2015), along with two label smoothing methods tailored for node classification, SALS (Wang et al., 2021) and ALS (Zhou et al., 2023).

**Results** In Table 1, the classification accuracy and 95% confidence interval for each of the seven models across the 10 datasets are presented. In most cases, PosteL outperforms baseline methods across various settings, demonstrating significant performance enhancements and validating its effectiveness for node classification. Specifically, our method performs better in 76 cases out of 80 settings against the ground truth labels. Furthermore, among these settings, 41 cases show improvements over the 95% confidence interval. Notably, on the Cornell dataset with the GCN backbone, our method achieves a substantial performance enhancement of 14.43%. When compared to the other soft label methods, PosteL performs better in most cases as well. The knowledge distillation method shows comparable performance with the GPR-GNN baseline, but even in this case, there are marginal differences between the two approaches. Our method outperforms SALS and ALS on both homophilic and heterophilic datasets. Specifically, our method demonstrates performance enhancement compared to SALS across all experimental settings and outperforms ALS in 71 out of

Table 1: Classification accuracy on 10 node classification datasets. Δ represents the performance improvement achieved by PosteL compared to the backbone model trained with the ground truth label. All results of the backbone model trained with the ground truth label are sourced from He et al. (2021).

| | Homophilic | | | | | Heterophilic | | | | |
|---|---|---|---|---|---|---|---|---|---|---|
| | Cora | CiteSeer | PubMed | Computers | Photo | Chameleon | Actor | Squirrel | Texas | Cornell |
| GCN | 87.14±1.01 | 79.86±0.67 | 86.74±0.27 | 83.32±0.33 | 88.26±0.73 | 59.61±2.21 | 33.23±1.16 | 46.78±0.87 | 77.38±3.28 | 65.90±4.43 |
| +LS | 87.77±0.97 | 81.06±0.59 | 87.73±0.24 | 89.08±0.30 | 94.05±0.26 | 64.81±1.53 | 33.81±0.75 | 49.53±1.10 | 77.87±3.11 | 67.87±3.77 |
| +KD | 87.90±0.90 | 80.97±0.56 | 87.03±0.29 | 88.56±0.36 | 93.64±0.31 | 64.49±1.38 | 33.33±0.78 | 49.38±0.64 | 78.03±2.62 | 63.61±5.57 |
| +SALS | 88.10±1.08 | 80.52±0.85 | 87.23±0.13 | 88.88±0.54 | 93.80±0.31 | 63.00±1.75 | 33.24±0.92 | 49.16±0.77 | 70.00±3.93 | 58.36±7.54 |
| +ALS | 88.10±0.85 | 81.02±0.52 | 87.30±0.30 | 89.18±0.36 | 93.88±0.27 | 64.11±1.29 | 34.05±0.49 | 47.44±0.76 | 77.38±2.13 | 71.64±3.28 |
| +PosteL | **88.56±0.90** | **82.10±0.50** | **88.00±0.25** | **89.30±0.23** | **94.08±0.35** | **65.80±1.23** | **35.16±0.43** | **52.76±0.64** | **80.82±2.79** | **80.33±1.80** |
| Δ | +1.42(↑) | +2.24(↑) | +1.26(↑) | +5.98(↑) | +5.82(↑) | +6.19(↑) | +1.93(↑) | +5.98(↑) | +3.44(↑) | +14.43(↑) |
| GAT | 88.03±0.79 | 80.52±0.71 | 87.04±0.24 | 83.32±0.39 | 90.94±0.68 | 63.13±1.93 | 33.93±2.47 | 44.49±0.88 | **80.82±2.13** | 78.21±2.95 |
| +LS | 88.69±0.99 | 81.27±0.86 | 86.33±0.32 | 88.95±0.15 | 94.06±0.39 | 65.16±1.49 | 34.55±1.15 | 45.94±1.60 | 78.69±4.10 | 74.10±4.10 |
| +KD | 87.47±0.94 | 80.79±0.60 | 86.54±0.31 | 88.99±0.46 | 93.76±0.31 | 65.14±1.47 | 35.13±1.36 | 43.86±0.85 | 79.02±2.46 | 73.44±2.46 |
| +SALS | 88.64±0.94 | 81.23±0.59 | 86.49±0.25 | 88.75±0.36 | 93.74±0.37 | 62.76±1.42 | 33.91±1.41 | 42.29±0.94 | 74.92±4.43 | 65.57±10.00 |
| +ALS | 88.60±0.92 | 81.09±0.68 | 87.06±0.24 | 89.57±0.35 | 94.16±0.36 | 66.15±1.25 | 34.05±0.52 | 46.85±1.45 | 78.03±3.11 | 75.08±3.77 |
| +PosteL | **89.21±1.08** | **82.13±0.64** | **87.08±0.19** | **89.60±0.29** | **94.31±0.31** | **66.28±1.14** | **35.92±0.72** | **49.38±1.05** | 80.33±2.62 | **80.33±1.81** |
| Δ | +1.18(↑) | +1.61(↑) | +0.04(↑) | +6.28(↑) | +3.37(↑) | +3.15(↑) | +1.99(↑) | +4.89(↑) | −0.49(↓) | +2.12(↑) |
| APPNP | 88.14±0.73 | 80.47±0.74 | 88.12±0.31 | 85.32±0.37 | 88.51±0.31 | 51.84±1.82 | 39.66±0.55 | 34.71±0.57 | 90.98±1.64 | 91.81±1.96 |
| +LS | 89.01±0.64 | 81.58±0.61 | 88.90±0.32 | 87.28±0.27 | 94.34±0.23 | **53.98±1.47** | 39.44±0.78 | **36.81±0.98** | 91.31±1.48 | 89.51±1.81 |
| +KD | 89.16±0.74 | 81.88±0.61 | 88.04±0.39 | 86.28±0.44 | 93.85±0.26 | 52.17±1.23 | **41.43±0.95** | 35.28±1.10 | 90.33±1.64 | 91.48±1.97 |
| +SALS | 88.97±0.90 | 81.53±0.56 | 88.50±0.31 | 86.49±0.50 | 93.74±0.38 | 52.82±1.95 | 39.66±0.64 | 36.34±0.65 | 83.44±3.93 | 89.51±3.77 |
| +ALS | 88.93±0.94 | 81.75±0.59 | **89.30±0.30** | 87.32±0.23 | 94.33±0.24 | 53.44±1.99 | 39.89±0.67 | 36.11±0.81 | 90.82±2.62 | 92.13±1.48 |
| +PosteL | **89.62±0.84** | **82.47±0.66** | 89.17±0.26 | **87.46±0.29** | **94.42±0.24** | 53.83±1.66 | 40.18±0.50 | 36.71±0.60 | **92.13±1.48** | **93.44±1.64** |
| Δ | +1.48(↑) | +2.00(↑) | +1.05(↑) | +2.14(↑) | +5.91(↑) | +1.99(↑) | +0.52(↑) | +2.00(↑) | +1.15(↑) | +1.63(↑) |
| MLP | 76.96±0.95 | 76.58±0.88 | 85.94±0.22 | 82.85±0.38 | 84.72±0.34 | 46.85±1.51 | 40.19±0.56 | 31.03±1.18 | 91.45±1.14 | 90.82±1.63 |
| +LS | 77.21±0.97 | 76.82±0.66 | 86.14±0.35 | 83.62±0.48 | 89.46±0.44 | 48.23±1.23 | 39.75±0.63 | 31.10±0.80 | 90.98±1.64 | 90.98±1.31 |
| +KD | 76.32±0.94 | 77.75±0.75 | 85.10±0.29 | 83.89±0.53 | 88.23±0.38 | 47.40±1.75 | **41.32±0.75** | 32.58±0.83 | 89.34±1.97 | 91.80±1.15 |
| +SALS | 77.29±1.05 | 77.00±0.90 | 85.78±0.33 | 82.55±0.51 | 89.11±0.52 | 43.68±1.69 | 39.47±0.73 | 30.88±0.68 | 86.39±5.09 | 89.11±0.52 |
| +ALS | 77.59±0.69 | 77.24±0.82 | 86.43±0.43 | **84.26±0.66** | 89.86±0.43 | 48.03±1.38 | 39.98±0.94 | 31.33±0.89 | 91.64±3.44 | 91.64±1.31 |
| +PosteL | **78.39±0.94** | **78.40±0.71** | **86.51±0.33** | 84.20±0.55 | **89.90±0.27** | **48.51±1.66** | 40.15±0.46 | **33.11±0.60** | **92.95±1.31** | **93.61±1.80** |
| Δ | +1.43(↑) | +1.82(↑) | +0.57(↑) | +1.35(↑) | +5.18(↑) | +1.66(↑) | −0.04(↓) | +2.08(↑) | +1.50(↑) | +2.79(↑) |
| ChebNet | 86.67±0.82 | 79.11±0.75 | 87.95±0.28 | 87.54±0.43 | 93.77±0.32 | 59.28±1.25 | 37.61±0.89 | 40.55±0.42 | 86.22±2.45 | 83.93±2.13 |
| +LS | 87.22±0.99 | 79.70±0.63 | 88.48±0.29 | 89.55±0.38 | 94.53±0.37 | 66.41±1.16 | 39.39±0.73 | 42.55±1.11 | **87.21±2.62** | 84.59±2.30 |
| +KD | 87.36±0.95 | 80.80±0.72 | 88.41±0.20 | 89.81±0.30 | 94.76±0.30 | 61.47±1.23 | **40.68±0.50** | 43.88±1.97 | 84.75±3.61 | 83.61±2.30 |
| +SALS | 87.31±0.94 | 79.71±0.83 | 88.46±0.30 | 89.52±0.35 | 94.19±0.27 | 56.94±2.52 | 39.25±0.67 | 41.61±0.93 | 74.26±3.61 | 73.44±6.89 |
| +ALS | 87.39±0.97 | 79.81±0.61 | 88.80±0.33 | 89.88±0.36 | **95.21±0.23** | 61.09±0.63 | 39.61±1.41 | 41.98±0.85 | 85.57±3.28 | 86.39±2.30 |
| +PosteL | **88.57±0.92** | **82.48±0.52** | **89.20±0.31** | **89.95±0.40** | 94.87±0.25 | **66.83±0.77** | 39.56±0.51 | **50.87±0.90** | 86.39±2.46 | **88.52±2.63** |
| Δ | +1.90(↑) | +3.37(↑) | +1.25(↑) | +2.41(↑) | +1.10(↑) | +7.55(↑) | +1.95(↑) | +10.32(↑) | +0.17(↑) | +4.59(↑) |
| GPR-GNN | 88.57±0.69 | 80.12±0.83 | 88.46±0.33 | 86.85±0.25 | 93.85±0.28 | 67.28±1.09 | 39.92±0.67 | 50.15±1.92 | 92.95±1.31 | 91.37±1.81 |
| +LS | 88.82±0.99 | 79.78±1.06 | 88.24±0.42 | 88.39±0.48 | 93.97±0.33 | 67.90±1.01 | 39.72±0.70 | 53.39±1.80 | 92.79±1.15 | 90.49±2.46 |
| +KD | **89.33±1.03** | **81.24±0.85** | 89.85±0.56 | 87.88±1.11 | 94.23±0.51 | 66.76±1.31 | **42.00±0.63** | 53.26±1.07 | **94.26±1.48** | 88.52±1.97 |
| +SALS | 88.78±0.90 | 80.71±0.91 | 90.12±0.46 | 88.63±0.35 | 94.23±0.65 | 65.16±1.49 | 39.67±0.73 | 44.75±1.45 | 73.61±3.44 | 82.46±2.95 |
| +ALS | 88.93±1.31 | 80.31±0.71 | 90.23±0.50 | 89.14±0.48 | 94.55±0.53 | 67.79±1.07 | 40.09±0.72 | 51.34±1.00 | 92.95±1.31 | 89.18±2.13 |
| +PosteL | 89.20±1.07 | 81.21±0.64 | **90.57±0.31** | **89.84±0.43** | **94.76±0.38** | **68.38±1.12** | 40.08±0.69 | **53.54±0.79** | 93.28±1.31 | **92.46±0.99** |
| Δ | +0.63(↑) | +1.09(↑) | +2.11(↑) | +2.99(↑) | +0.91(↑) | +1.10(↑) | +0.16(↑) | +3.39(↑) | +0.33(↑) | +1.09(↑) |
| BernNet | 88.52±0.95 | 80.09±0.79 | 88.48±0.41 | 87.64±0.44 | 93.63±0.35 | 68.29±1.58 | **41.79±1.01** | 51.35±0.73 | 93.12±0.65 | 92.13±1.64 |
| +LS | 88.80±0.92 | 80.37±1.05 | 87.40±0.27 | 88.32±0.38 | 93.70±0.21 | 69.58±0.94 | 39.60±0.53 | 52.39±0.60 | 91.80±1.80 | 90.49±1.48 |
| +KD | 87.78±0.99 | 81.20±0.86 | 87.59±0.41 | 87.35±0.40 | 93.96±0.40 | 67.75±1.42 | 41.04±0.89 | 51.25±0.83 | 93.61±1.31 | 90.33±2.30 |
| +SALS | 88.77±0.85 | 81.20±0.61 | 88.61±0.35 | 88.87±0.33 | 94.22±0.43 | 64.62±0.85 | 40.15±1.07 | 46.19±0.78 | 85.90±4.10 | 88.03±3.12 |
| +ALS | 89.13±0.94 | 81.17±0.60 | **89.30±0.46** | 89.52±0.30 | **94.54±0.32** | 67.92±1.07 | 40.51±0.61 | 51.83±1.31 | 93.77±1.31 | 92.79±1.48 |
| +PosteL | **89.39±0.92** | **82.46±0.67** | 89.07±0.29 | **89.56±0.35** | **94.54±0.36** | **69.65±0.83** | 40.40±0.67 | **53.11±0.87** | **93.93±1.15** | **92.95±1.80** |
| Δ | +0.87(↑) | +2.37(↑) | +0.59(↑) | +1.92(↑) | +0.91(↑) | +1.36(↑) | −1.39(↓) | +1.76(↑) | +0.81(↑) | +0.82(↑) |
| OrderedGNN | 88.62±1.05 | 80.11±0.86 | 88.74±0.56 | 89.72±0.50 | 94.76±0.36 | 58.27±1.33 | 39.73±1.15 | 38.70±1.10 | 90.16±2.63 | 90.33±2.46 |
| +LS | 88.52±0.94 | 80.23±0.80 | 88.16±0.33 | 89.59±0.47 | 94.49±0.45 | 58.86±1.62 | 40.01±0.66 | 40.12±0.82 | 88.20±3.61 | 91.15±1.31 |
| +KD | 88.26±1.07 | 80.52±0.83 | 88.23±0.21 | 89.35±0.34 | 94.40±0.23 | 58.21±1.18 | 40.17±0.45 | 40.92±0.87 | **90.49±1.48** | 91.31±1.80 |
| +SALS | 88.44±0.97 | 80.93±0.72 | 88.08±0.62 | 88.94±0.51 | 93.87±0.35 | 59.30±1.25 | 39.52±0.41 | 40.85±0.86 | 77.70±4.75 | 84.75±4.10 |
| +ALS | 87.96±0.74 | 80.60±0.57 | 89.09±0.57 | 89.84±0.48 | 94.76±0.36 | 59.39±1.23 | 40.28±0.79 | 40.37±1.05 | 90.00±2.62 | 89.84±2.95 |
| +PosteL | **88.97±1.15** | **82.54±0.64** | **88.85±0.61** | **90.13±0.29** | **94.96±0.34** | **60.15±1.20** | 39.99±1.00 | **43.72±0.85** | 87.70±5.25 | **91.97±1.15** |
| Δ | +0.35(↑) | +2.43(↑) | +0.11(↑) | +0.41(↑) | +0.20(↑) | +1.88(↑) | +0.26(↑) | +5.02(↑) | −2.46(↓) | +1.64(↑) |

80 settings. Especially, we observe a significant performance gap on heterophilic datasets, which aligns with our assumption that label smoothing methods relying on the homophilic assumption should harm training for heterophilic datasets.

## 4.2 ANALYSIS

In this section, we analyze the main experimental result from various perspectives, including design choices, ablations, and computational complexity.

**Loss curves analysis** We investigate the influence of soft labels on the learning dynamics of GNNs by visualizing the loss function of GCNs with and without soft labels. Figure 2 visualizes the differ-

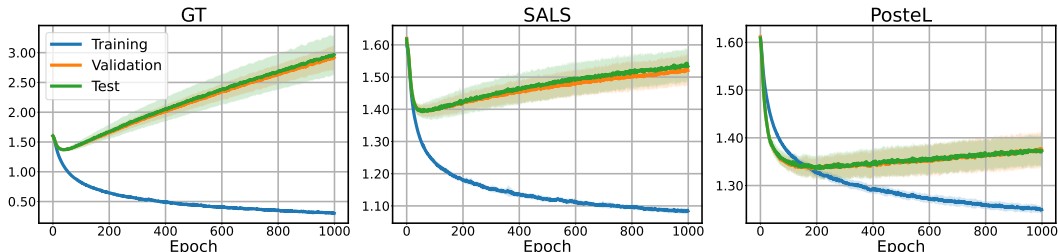

Figure 2: Loss curve of GCN trained on PosteL, SALS, and ground truth labels on the Squirrel dataset.

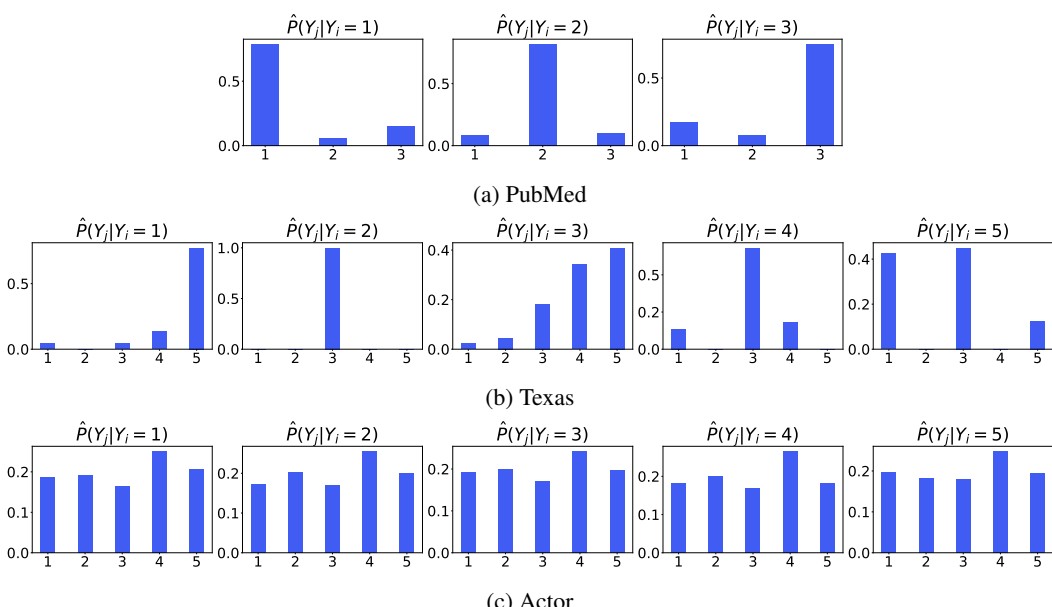

(a) PubMed

(b) Texas

(c) Actor

Figure 3: Empirical conditional distributions between two adjacent nodes. We omit the adjacent condition $(i, j) \in \mathcal{E}$ from the figures for simplicity.

ences between training, validation, and test losses with ground truth labels, SALS labels, and PosteL labels on the Squirrel dataset. We observe that the gap between the training and the validation or test loss of PosteL is smaller than that of other baselines. Furthermore, while other baselines exhibit strong overfitting after 50 epochs, PosteL shows no signs of overfitting even up to 200 epochs. We conjecture that predicting the correct PosteL label implies the correct prediction of the local neighborhood structure since the PosteL labels contain the local neighborhood information of the target node. Hence, the model trained with PosteL labels could have a better understanding of the graph structure, potentially leading to a better generalization performance. A similar context prediction approach has been proposed as a pertaining method in previous studies (Hu et al., 2019; Rong et al., 2020). We provide the same curves for all datasets in Figure 8 and Figure 9 in Appendix H. All curves across all datasets show similar patterns.

**Influence of neighborhood label distribution** Our approach assumes that the distribution of neighborhood labels varies depending on the label of the target node. If there are no significant differences between the neighborhood's label distributions, the posterior relabeling assigns similar soft labels for all nodes, making our method similar to the uniform noise method.

Figure 3 shows the neighborhood label distribution for three different datasets. In the PubMed and Texas datasets, we observe a notable difference in the conditionals w.r.t the different labels of a target node. The PubMed dataset is known to be homophilic, where nodes with the same labels are likely to be connected, and the conditional distributions match the characteristics of the homophilic

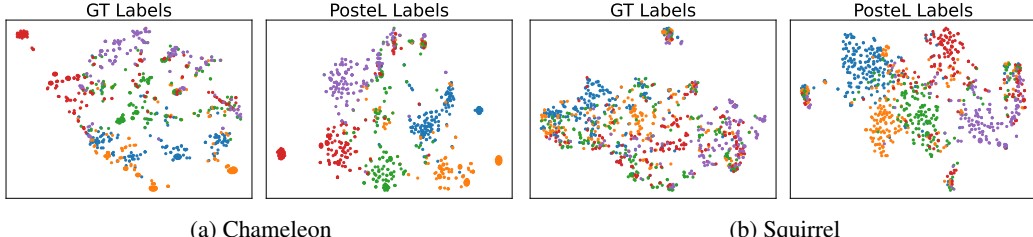

(a) Chameleon                           (b) Squirrel

Figure 4: t-SNE plots of the final layer representation of the Chameleon and Squirrel datasets. For each dataset, the left figure displays the representations trained on the ground truth labels, while the right figure displays the representations trained on the PosteL labels.

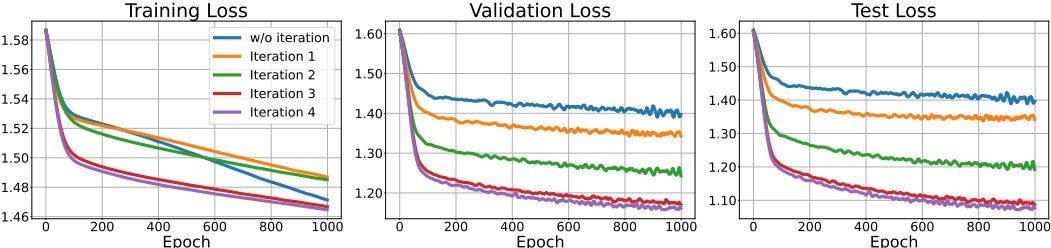

Figure 5: The impact of the iterative pseudo labeling: loss curves of GCN on the Cornell dataset.

dataset. The Texas dataset, a heterophilic dataset, shows that some pairs of labels more frequently appear in the graph. For example, when the target node has the label of 1, their neighborhoods will likely have the label of 5. On the other hand, the conditionals of the Actor dataset do not vary much regarding the label of the target node. In such a case, the prior will likely dominate the posterior. Therefore, the posterior may not provide useful information about neighborhood nodes, potentially limiting the effectiveness of our method. This analysis aligns with the results in Table 1, where the improvement of the Actor dataset is less significant than those of the PubMed and Texas datasets. The neighborhood label distributions for all datasets are provided in Figure 10 and Figure 11 in Appendix I.

**Visualization of node embeddings**     Figure 4 presents the t-SNE (Van der Maaten & Hinton, 2008) plots of node embeddings from the GCN with the Chameleon and Squirrel datasets. The node color represents the label. For each dataset, the left plot visualizes the embeddings with the ground truth labels, while the right plot visualizes the embeddings with PosteL labels. The visualization shows that the embeddings from the soft labels form tighter clusters compared to those trained with the ground truth labels. This visualization results coincide with the t-SNE visualization of the previous work of Müller et al. (2019).

**Effect of iterative pseudo labeling**     We evaluate the impact of iterative pseudo labeling by analyzing the loss curve at each iteration. Figure 5 illustrates the loss curves for different iterations on the Cornell dataset. As the iteration progresses, the validation and test losses after 1,000 epochs keep decreasing. In this example, the model performs best after four iteration steps. We find that the best validation performance is obtained from 1.13 iterations on average. We provide the average iteration steps in Appendix F used to report the results in Table 1.

**Design choices of likelihood model**     We explore various valid design choices for likelihood models. We introduce two variants of PosteL: PosteL (normalized) and PosteL (local-$H$). In Equation (2), each edge has an equal contribution to the conditional. The conditional can be influenced by a few numbers of nodes with many connections. To reduce dependency on high-degree nodes, we alternatively test the following conditional, denoted as PosteL (normalized):

$$\hat{P}^{\text{norm.}}(Y_j = m | Y_i = n, (i,j) \in \mathcal{E}) := \frac{\sum_{y_u = n} \sum_{v \in \mathcal{N}(u)} \frac{1}{|\mathcal{N}(u)|} \cdot \mathbb{1}[y_v = m]}{|\{y_u = n \mid u \in \mathcal{V}\}|},$$

Table 2: Classification accuracy with various choices of likelihood model. PosteL (local-1) and (local-2) indicate that the likelihood is estimated within one- and two-hop neighbors of a target node, respectively. PosteL (norm.), shortened from PosteL (normalized), indicates that the likelihood is normalized based on the degree of a node.

|  | Cora | CiteSeer | Computers | Photo | Chameleon | Actor | Texas | Cornell |
|---|---|---|---|---|---|---|---|---|
| GCN | $87.14_{\pm1.01}$ | $79.86_{\pm0.67}$ | $83.32_{\pm0.33}$ | $88.26_{\pm0.73}$ | $59.61_{\pm2.21}$ | $33.23_{\pm1.16}$ | $77.38_{\pm3.28}$ | $65.90_{\pm4.43}$ |
| +PosteL (local-1) | $88.26_{\pm1.07}$ | $81.42_{\pm0.46}$ | $89.08_{\pm0.31}$ | $93.61_{\pm0.40}$ | $65.36_{\pm1.25}$ | $33.48_{\pm1.03}$ | $79.02_{\pm3.11}$ | $71.97_{\pm4.10}$ |
| +PosteL (local-2) | $88.62_{\pm0.97}$ | $81.92_{\pm0.42}$ | $88.62_{\pm0.48}$ | $93.95_{\pm0.37}$ | $65.10_{\pm1.55}$ | $34.63_{\pm0.46}$ | $78.20_{\pm2.79}$ | $73.28_{\pm4.10}$ |
| +PosteL (norm.) | $\mathbf{89.00_{\pm0.99}}$ | $81.86_{\pm0.70}$ | $\mathbf{89.30_{\pm0.39}}$ | $\mathbf{94.13_{\pm0.39}}$ | $\mathbf{66.00_{\pm1.14}}$ | $34.90_{\pm0.63}$ | $80.33_{\pm2.95}$ | $80.00_{\pm1.97}$ |
| +PosteL | $88.56_{\pm0.90}$ | $\mathbf{82.10_{\pm0.50}}$ | $\mathbf{89.30_{\pm0.23}}$ | $94.08_{\pm0.35}$ | $65.80_{\pm1.23}$ | $\mathbf{35.16_{\pm0.43}}$ | $\mathbf{80.82_{\pm2.79}}$ | $\mathbf{80.33_{\pm1.80}}$ |

Table 3: Ablation studies on three main components of PosteL on GCN. PS stands for posterior label smoothing without uniform noise, UN stands for uniform noise added to the posterior distribution, and IPL stands for iterative pseudo labeling. We use ✓ to indicate the presence of the corresponding component in training and ✗ to indicate its absence. IPL with one indicates the performance with a single pseudo labeling step.

| PS | UN | IPL | Cora | CiteSeer | Computers | Photo | Chameleon | Actor | Texas | Cornell |
|---|---|---|---|---|---|---|---|---|---|---|
| ✗ | ✗ | ✗ | $87.14_{\pm1.01}$ | $79.86_{\pm0.67}$ | $83.32_{\pm0.33}$ | $88.26_{\pm0.73}$ | $59.61_{\pm2.21}$ | $33.23_{\pm1.16}$ | $77.38_{\pm3.28}$ | $65.90_{\pm4.43}$ |
| ✓ | ✗ | ✗ | $88.11_{\pm1.22}$ | $80.95_{\pm0.52}$ | $88.86_{\pm0.40}$ | $93.55_{\pm0.30}$ | $64.53_{\pm1.23}$ | $33.48_{\pm0.62}$ | $78.52_{\pm2.46}$ | $68.52_{\pm4.43}$ |
| ✗ | ✓ | ✗ | $87.77_{\pm0.97}$ | $81.06_{\pm0.59}$ | $89.08_{\pm0.30}$ | $94.05_{\pm0.26}$ | $64.81_{\pm1.53}$ | $33.81_{\pm0.75}$ | $77.87_{\pm3.11}$ | $67.87_{\pm3.77}$ |
| ✓ | ✗ | ✓ | $\mathbf{88.56_{\pm0.90}}$ | $81.64_{\pm0.57}$ | $88.70_{\pm0.27}$ | $93.70_{\pm0.37}$ | $64.25_{\pm1.93}$ | $34.71_{\pm0.76}$ | $\mathbf{80.82_{\pm2.79}}$ | $80.16_{\pm1.97}$ |
| ✓ | ✓ | ✗ | $87.83_{\pm0.92}$ | $82.09_{\pm0.44}$ | $89.17_{\pm0.31}$ | $93.98_{\pm0.34}$ | $\mathbf{66.19_{\pm1.60}}$ | $34.91_{\pm0.48}$ | $79.51_{\pm3.61}$ | $71.97_{\pm5.25}$ |
| ✓ | ✓ | 1 | $87.96_{\pm0.90}$ | $\mathbf{82.33_{\pm0.52}}$ | $89.16_{\pm0.30}$ | $94.06_{\pm0.27}$ | $65.89_{\pm1.51}$ | $34.96_{\pm0.48}$ | $80.16_{\pm2.79}$ | $\mathbf{80.33_{\pm1.97}}$ |
| ✓ | ✓ | ✓ | $\mathbf{88.56_{\pm0.90}}$ | $82.10_{\pm0.50}$ | $\mathbf{89.30_{\pm0.23}}$ | $\mathbf{94.08_{\pm0.35}}$ | $65.80_{\pm1.23}$ | $\mathbf{35.16_{\pm0.43}}$ | $\mathbf{80.82_{\pm2.79}}$ | $\mathbf{80.33_{\pm1.80}}$ |

where $\mathbb{1}$ is an indicator function.

In PosteL (local-$H$), we estimate the likelihood and prior distributions of each node from their respective $H$-hop ego graphs. Specifically, the likelihood of PosteL (local-$H$) is formulated as follows:

$$\hat{P}^{\text{local-}H}(Y_j = m|Y_i = n, (i,j) \in \mathcal{E}) := \frac{|\{(u,v)|y_v = m, y_u = n, (u,v) \in \mathcal{E}, u,v \in \mathcal{N}^{(H)}(i)\}|}{|\{(u,v)|y_u = n, (u,v) \in \mathcal{E}, u,v \in \mathcal{N}^{(H)}(i)\}|},$$

where $\mathcal{N}^{(H)}(i)$ denotes the set of neighborhoods of node $i$ within $H$ hops. Through the local likelihood, we test the importance of global and local statistics in the smoothing process.

Table 2 shows the comparison between these variants. The likelihood with global statistics, e.g., PosteL and PosteL (normalized), performs better than the local likelihood methods, e.g., PosteL (local-1) and PosteL (local-2) in general, highlighting the importance of simultaneously utilizing global statistics. Especially in the Cornell dataset, a significant performance gap between PosteL and PosteL (local) is observed. PosteL (normalized) demonstrates similar performance to PosteL.

**Ablation studies** To highlight the importance of each component in PosteL, we perform ablation studies on three components: posterior smoothing without uniform noise (PS), uniform smoothing (UN), and iterative pseudo labeling (IPL). Table 3 presents the performance results from the ablation studies.

The configuration with all components included achieves the highest performance, underscoring the significance of each component. The iterative pseudo labeling proves effective across almost all datasets, with a particularly notable impact on the Cornell dataset. However, even without iterative pseudo labeling, the performance remains competitive, suggesting that its use can be decided based on available resources. Additionally, incorporating uniform noise into the posterior distribution enhances performance on several datasets. Moreover, PosteL consistently outperforms the approach using only uniform noise, a widely used label smoothing method.

**Complexity analysis** The computational complexity of calculating the posterior label is $O(|\mathcal{E}|K)$. Since the labeling is performed before the learning stage, the time required to process the posterior label can be considered negligible. The training time increases linearly w.r.t the number of iterations

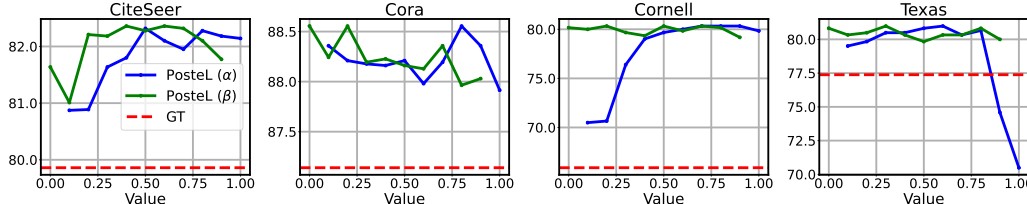

Figure 6: Hyperparameter sensitivity analysis on GCN.

Table 4: Accuracy of the model trained with sparse labels. The ratio indicates the percentage of nodes used for training.

| | ratio | Cora | CiteSeer | Computers | Photo | Chameleon | Actor | Texas | Cornell |
|---|---|---|---|---|---|---|---|---|---|
| GCN | | 80.66±0.89 | 73.52±1.43 | 84.47±0.99 | 92.38±0.41 | 45.01±3.52 | 24.62±5.83 | 67.05±14.92 | 58.36±19.19 |
| +LS | | 80.72±0.95 | 73.48±1.71 | 85.32±0.68 | 92.82±0.39 | 47.61±2.91 | 27.59±2.52 | 69.34±14.92 | 59.34±16.23 |
| +SALS | 10% | 81.20±0.95 | 75.48±1.20 | **85.92±0.84** | 92.59±0.38 | 46.11±2.56 | 28.81±2.01 | 63.44±13.93 | 58.69±14.93 |
| +ALS | | 80.97±0.89 | 74.02±1.54 | 85.24±0.79 | 92.87±0.34 | 45.49±3.09 | 27.59±2.13 | 67.87±14.26 | 61.48±15.57 |
| +PosteL | | **82.33±1.28** | **76.15±1.05** | 85.50±0.50 | **92.99±0.31** | **51.49±2.28** | **31.25±2.59** | **71.48±13.93** | **67.54±16.40** |
| GCN | | 82.91±0.94 | 75.91±1.20 | 86.75±0.36 | 92.99±0.32 | 52.67±1.51 | 30.18±1.51 | 65.90±14.92 | 55.25±9.68 |
| +LS | | 83.07±1.05 | 76.03±0.93 | 87.00±0.41 | 93.26±0.36 | 53.89±1.49 | 29.49±1.39 | 71.15±7.70 | 56.56±11.15 |
| +SALS | 20% | 84.25±1.30 | 77.09±1.02 | **87.23±0.39** | 93.10±0.34 | 54.60±2.04 | 29.90±1.36 | 64.43±11.64 | 52.62±13.45 |
| +ALS | | 83.25±1.07 | 76.40±1.09 | 86.87±0.49 | 93.36±0.34 | 53.28±1.29 | 30.49±1.57 | 66.56±15.25 | 62.46±10.66 |
| +PosteL | | **85.17±1.02** | **79.36±0.61** | **87.23±0.30** | **93.40±0.35** | **56.81±0.90** | **32.91±1.51** | **72.13±6.72** | **79.84±1.97** |

with the pseudo labeling. However, experiments show that an average of 1.13 iterations is needed, making our approach feasible without having too many iterations. The proof of computational complexity is in Appendix F.

**Hyperparameter sensitivity analysis** Figure 6 shows the performance with varying values of $\alpha$ and $\beta$ on GCN. The blue line indicates the performance with varying $\alpha$, and the green line shows the performance with varying $\beta$. The red dotted line represents the performance with the ground truth label. Regardless of the values of $\alpha$ and $\beta$, the performance consistently outperforms the case using ground truth labels, indicating that PosteL is insensitive to $\alpha$ and $\beta$. We observe that $\alpha$ values greater than 0.8 may harm training, suggesting the necessity of interpolating ground truth labels.

### 4.3 TRAINING WITH SPARSE LABELS

Our method relies on global statistics estimated from training nodes. However, in scenarios where training data is sparse, the estimation of global statistics can be challenging. To assess the effectiveness of the label smoothing from graphs with sparse labels, we conduct experiments with varying sizes of a training set. We conduct the classification experiments with the same settings as in the previous section, but only used 10% to 20% of the training nodes defined in that section. The percentage of validation and test nodes is set to 20% for all experiments. Table 4 provides the classification performance with sparse labels. Even in scenarios with sparse labels, PosteL consistently outperforms models trained on ground truth labels in most cases. These results show that our method can effectively capture global statistics even when training data is limited. We provide additional experiments on extremely sparse labels in Appendix G.

## 5 CONCLUSION

In this paper, we proposed a novel posterior label smoothing method, PosteL, designed to enhance node classification performance in graph-structured data. Our approach integrates both local neighborhood information and global label statistics to generate soft labels, thereby improving generalization and mitigating overfitting. Extensive experiments across various datasets and models demonstrated the effectiveness of PosteL, showing significant performance gains compared to baseline methods despite its simplicity.

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

# A ANALYSIS OF ASSUMPTIONS AND CHARACTERISTICS OF POSTEL IN HETEROPHILIC GRAPHS

## A.1 IN-DEPTH ANALYSIS OF THE UNDERLYING ASSUMPTIONS OF POSTEL

Our posterior distribution, $P(Y_i = k | \{Y_j = y_j\}_{j \in \mathcal{N}(i)})$, is based on the assumption that nodes with similar neighborhood label distributions should exhibit similar characteristics. Xiao et al. (2024) introduce neighborhood context similarity, $\mathcal{S}(\mathcal{G})$, defined as:

$$\mathcal{S}(\mathcal{G}) = \sum_{k=1}^{K} \sum_{u,v \in \mathcal{V}_\parallel} \cos(d(u), d(v)), \tag{4}$$

where $\mathcal{V}_k$ is the set of nodes with label $k$, $d(u)$ is the neighborhood label distribution, and $\cos(\cdot)$ is the cosine similarity. (We omit some terms for simplicity.) $\mathcal{S}(\mathcal{G})$ represents the degree to which the neighborhood distributions between nodes with the same label are similar. This metric is closely related to our assumption: if $\mathcal{S}(\mathcal{G})$ is large, our assumption holds. The table below, from Xiao et al. (2024), presents the edge homophily ratio $\mathcal{H}(\mathcal{G})$ and neighborhood context similarity $\mathcal{S}(\mathcal{G})$ for each dataset.

Table 5: The edge homophily ratio, $\mathcal{H}(\mathcal{G})$, and neighborhood context similarity, $\mathcal{S}(\mathcal{G})$, of the node classification datasets

|  | Cora | CiteSeer | PubMed | Computers | Photo | Chameleon | Actor | Squirrel | Texas | Cornell |
|---|---|---|---|---|---|---|---|---|---|---|
| $\mathcal{H}(\mathcal{G})$ | 0.81 | 0.74 | 0.80 | 0.78 | 0.83 | 0.23 | 0.22 | 0.22 | 0.11 | 0.30 |
| $\mathcal{S}(\mathcal{G})$ | 0.89 | 0.81 | 0.87 | 0.90 | 0.91 | 0.67 | 0.68 | 0.73 | 0.79 | 0.40 |

From this table, we observe that the homophilic assumption holds well on homophilic graphs but becomes fragile on heterophilic graphs. In contrast, our assumption is hold across both graph types. Therefore, our assumption can be considered more general. Therefore, our assumption can be considered more general. We conjecture that this generality contributes to PosteL's strong performance on both homophilic and heterophilic datasets.

## A.2 THEORETICAL ANALYSIS OF THE CHARACTERISTICS OF POSTEL WITH HETEROPHILIC GRAPHS

In this subsection, we provide two additional lemmas showing the characteristics of PosteL with heterophilic graphs. For simplicity, we focus on a binary classification problem with two classes: 0 and 1. To establish the theorem, we define the individual node homophily $p_i$ and class homophily $c_k$ as follows:

$$p_i := \frac{|\{(i,j)|(i,j) \in \mathcal{E}, y_i = y_j\}|}{|\{(i,j)|(i,j) \in \mathcal{E}\}|}, c_k := \frac{|\{(i,j)|(i,j) \in \mathcal{E}, y_i = k, y_j = k\}|}{|\{(i,j)|(i,j) \in \mathcal{E}, y_i = k\}|}. \tag{5}$$

Then the posterior label of the node being labeled to 0 is:

$$P(Y_i = 0 | \{Y_j = y_j\}_{j \in \mathcal{N}(i)}) = \frac{c_0^{|\mathcal{N}(i)|p_i}(1-c_0)^{|\mathcal{N}(i)|(1-p_i)}}{c_0^{|\mathcal{N}(i)|p_i}(1-c_0)^{|\mathcal{N}(i)|(1-p_i)} + c_1^{|\mathcal{N}(i)|(1-p_i)}(1-c_1)^{|\mathcal{N}(i)|p_i}}. \tag{6}$$

**Lemma A.1.** *In a heterophilic graph with $c_0, c_1 < 0.5$, if two nodes have the same degree $d$, and node $i$ is connected to more nodes labeled 1 than node $j$, then PosteL assigns a higher probability of node $i$ being labeled 0 compared to node $j$.*

*Proof.* Since the number of adjacent nodes with a different label is larger for node $i$, we have $p_i < p_j$. The lemma can be expressed as follows:

$$\frac{c_0^{dp_i}(1-c_0)^{d(1-p_i)}}{c_0^{dp_i}(1-c_0)^{d(1-p_i)}+c_1^{d(1-p_i)}(1-c_1)^{dp_i}} > \frac{c_0^{dp_j}(1-c_0)^{d(1-p_j)}}{c_0^{dp_j}(1-c_0)^{d(1-p_j)}+c_1^{d(1-p_j)}(1-c_1)^{dp_j}}. \quad (7)$$

Expanding the inequality:

$$c_0^{d(p_i+p_j)}(1-c_0)^{d(2-p_i-p_j)} + c_0^{dp_i}(1-c_0)^{d(1-p_i)}c_1^{d(1-p_j)}(1-c_1)^{dp_j} \quad (8)$$

$$> c_0^{d(p_i+p_j)}(1-c_0)^{d(2-p_i-p_j)} + c_0^{dp_j}(1-c_0)^{d(1-p_j)}c_1^{d(1-p_i)}(1-c_1)^{dp_i}. \quad (9)$$

Subtracting $c_0^{d(p_i+p_j)}(1-c_0)^{d(2-p_i-p_j)}$ from both sides:

$$c_0^{dp_i}(1-c_0)^{d(1-p_i)}c_1^{d(1-p_j)}(1-c_1)^{dp_j} > c_0^{dp_j}(1-c_0)^{d(1-p_j)}c_1^{d(1-p_i)}(1-c_1)^{dp_i}. \quad (10)$$

$$((1-c_0)(1-c_1))^{d(p_j-p_i)} > (c_0 c_1)^{d(p_j-p_i)}. \quad (11)$$

Since $c_0 < 1 - c_0$ and $c_1 < 1 - c_1$ imply that $(1-c_0)(1-c_1) > c_0 c_1$, and given $(1-c_0)(1-c_1) > c_0 c_1$ and $p_j - p_i > 0$, the inequality holds. $\square$

**Lemma A.2.** *In a heterophilic graph with $c_0, c_1 < 0.5$, if two nodes are connected only to nodes labeled 1, and their respective degrees are $n$ and $m$ ($n > m$), then PosteL assigns a higher probability of being labeled 0 to the node with the higher degree.*

*Proof.* When nodes are connected only to nodes labeled 1, PosteL assigns the posterior probability of the node being labeled 0 as follows:

$$P(Y_i = 0 | \{Y_j = 1\}_{j \in \mathcal{N}(i)}) = \frac{(1-c_0)^{\deg(i)}}{(1-c_0)^{\deg(i)} + c_1^{\deg(i)}}, \quad (12)$$

where $\deg(i)$ represents the degree of node $i$. The lemma can be expressed as follows:

$$\frac{(1-c_0)^n}{(1-c_0)^n + c_1^n} > \frac{(1-c_0)^m}{(1-c_0)^m + c_1^m}. \quad (13)$$

Expanding the inequality:

$$((1-c_0)^m + c_1^m)(1-c_0)^n > ((1-c_0)^n + c_1^n)(1-c_0)^m. \quad (14)$$

$$(1-c_0)^n(1-c_0)^m + c_1^m(1-c_0)^n > (1-c_0)^n(1-c_0)^m + c_1^n(1-c_0)^m. \quad (15)$$

$$c_1^m(1-c_0)^n > c_1^n(1-c_0)^m. \quad (16)$$

$$(1-c_0)^{n-m} > c_1^{n-m}. \quad (17)$$

Since $c_0, c_1 < 0.5$, it follows that $1 - c_0 > c_1$. Thus, the inequality holds. $\square$

As a corollary we can show the following two properties.

- If two nodes have the same degree, the node connected to more nodes labeled 1 should have a higher probability of being labeled 0.
- If two nodes are connected only to nodes labeled 1, the node with the higher degree should have a higher probability of being labeled 0.

Although these lemmas and corollary may not reflect the real-world scenario, analyzing properties of a model is an important step towards understanding its performance.

# B  ALGORITHMS RELATED TO ITERATIVE PSEUDO LABELING

Algorithm 2 and Algorithm 3 present the detailed algorithms for PosteL using pseudo labels and the training process involving iterative pseudo labeling.

---

**Algorithm 2** Posterior label smoothing using pseudo labels

---

**Require:** The set of training nodes $\mathcal{V}_{\text{train}}$ and the set of nodes with pseudo label $\mathcal{V}_{\text{pseudo}}$; the number of classes $K$; one-hot encoding of node labels $\{e_i\}_{i \in \mathcal{V}_{\text{train}} \cup \mathcal{V}_{\text{pseudo}}}$; and the hyperparameters $\alpha$ and $\beta$.

**Ensure:** The set of soft labels $\{\hat{e}_i\}_{i \in \mathcal{V}_{\text{train}}}$.

Initialize the set of labeled nodes: $\mathcal{V}_{\text{labeled}} = \mathcal{V}_{\text{train}} \cup \mathcal{V}_{\text{pseudo}}$

Estimate prior distribution for $m \in [K]$: $\hat{P}(Y_i = m) = \sum_{u \in \mathcal{V}_{\text{labeled}}} e_{um} / |\mathcal{V}_{\text{labeled}}|$.

Define the set of labeled neighbors for each node $u$: $\mathcal{N}_{\text{labeled}}(u) = \mathcal{N}(u) \cap \mathcal{V}_{\text{labeled}}$.

Estimate the empirical conditional for $n, m \in [K]$:

$$\hat{P}(Y_j = m | Y_i = n, (i,j) \in \mathcal{E}) \propto \sum_{u:u \in \mathcal{V}_{\text{labeled}}, y_u = n} \sum_{v \in \mathcal{N}_{\text{labeled}}(u)} e_{vm}.$$

**for** each $i \in \mathcal{V}_{\text{train}}$ such that $\mathcal{N}_{\text{labeled}}(i) \neq \emptyset$ **do**

  Approximate likelihood:

  $$P(\{Y_j = y_j\}_{j \in \mathcal{N}_{\text{labeled}}(i)} | Y_i = k) \approx \prod_{j \in \mathcal{N}_{\text{labeled}}(i)} \hat{P}(Y_j = y_j | Y_i = k, (i,j) \in \mathcal{E}).$$

  Compute posterior distribution: $P(Y_i = k \mid \{Y_j = y_j\}_{j \in \mathcal{N}_{\text{labeled}}(i)})$ using Equation (1).

  Add uniform noise: $\tilde{e}_{ik} \propto P(Y_i = k \mid \{Y_j = y_j\}_{j \in \mathcal{N}_{\text{labeled}}(i)}) + \beta \epsilon$.

  Obtain soft label: $\hat{e}_i = \alpha \tilde{e}_i + (1 - \alpha) e_i$.

**end for**

---

**Algorithm 3** Training the GNN with PosteL involving iterative pseudo labeling

---

**Require:** The input graph $\mathcal{G} = (\mathcal{V}, \mathcal{E}, \boldsymbol{X})$; the set of training nodes $\mathcal{V}_{\text{train}}$ and test nodes $\mathcal{V}_{\text{test}}$, where $\mathcal{V}_{\text{train}} \cup \mathcal{V}_{\text{test}} = \mathcal{V}$; one-hot encoded training labels $\{e_i\}_{i \in \mathcal{V}_{\text{train}}}$; and PosteL, as described in Algorithm 2, along with its parameters $K, \alpha$, and $\beta$.

**Ensure:** Trained GNN model $f$ with pseudo labeled nodes.

Initialize the pseudo labeled node set: $\mathcal{V}_{\text{pseudo}} = \emptyset$.

Initialize pseudo labels: $\{e_i\}_{i \in \mathcal{V}_{\text{pseudo}}} = \emptyset$.

**while** validation loss is decreasing **do**

  Apply posterior label smoothing:

  $$\{\hat{e}_i\}_{i \in \mathcal{V}_{\text{train}}} = \text{PosteL}(\mathcal{V}_{\text{train}}, \mathcal{V}_{\text{pseudo}}, \{e_i\}_{i \in \mathcal{V}_{\text{training}} \cup \mathcal{V}_{\text{pseudo}}}, K, \alpha, \beta).$$

  Train the GNN model $f$ to predict soft labels for the training nodes $\{\hat{e}_i\}_{i \in \mathcal{V}_{\text{train}}}$.

  Obtain pseudo labels $\{\bar{y}_i\}_{i \in \mathcal{V}_{\text{test}}}$ and their one-hot encodings $\{\bar{e}_i\}_{i \in \mathcal{V}_{\text{test}}}$ for test nodes:

  $$\{\bar{y}_i\}_{i \in \mathcal{V}_{\text{test}}} = \{\arg \max f(\mathcal{G})_i\}_{i \in \mathcal{V}_{\text{test}}}.$$

  Update the pseudo labeled node set: $\mathcal{V}_{\text{pseudo}} = \mathcal{V}_{\text{test}}$.

  Update pseudo labels: $\{e_i\}_{i \in \mathcal{V}_{\text{pseudo}}} = \{\bar{e}_i\}_{i \in \mathcal{V}_{\text{test}}}$.

**end while**

---

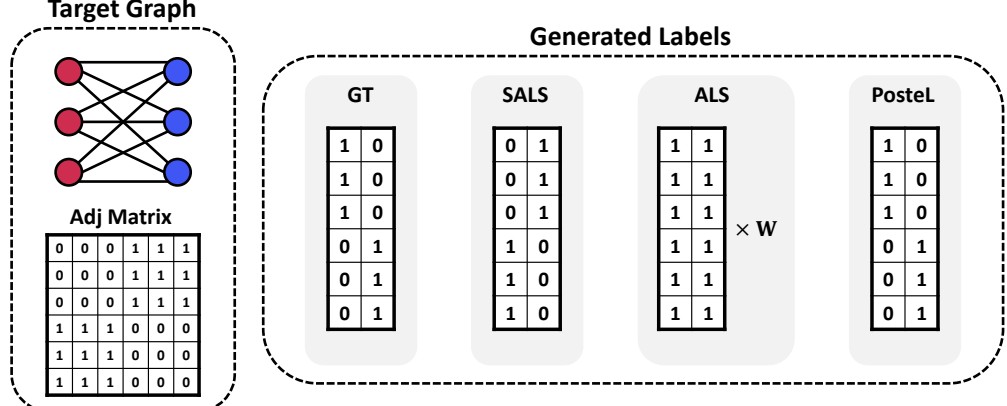

Figure 7: The toy example of the soft labels on a binary node classification task with a bipartite graph

## C    A CASE STUDY COMPARING SMOOTHING METHODS AND POSTEL

In this section, we aim to provide an in-depth explanation of the main differences between PosteL and other label smoothing methods to offer insight into why PosteL performs well, especially on heterophilic graphs.

### C.1    THE EFFECT OF UTILIZING GLOBAL STATISTICS

PosteL leverages global statistics, specifically $\hat{P}(Y_j = m|Y_i = n, (i,j) \in \mathcal{E})$ and $\hat{P}(Y_i = m)$, to generate soft labels. In contrast, SALS (Wang et al., 2021) only utilizes information from 1-hop neighbors. The use of global statistics in ALS (Zhou et al., 2023) is questionable due to the presence of learnable component in their soft label. Figure 7 shows an example of the soft labels on a binary node classification task with a bipartite graph. The toy example highlights the key differences between existing methods and ours, which will be elaborated further below.

**Conditional probability and its impact**    $\hat{P}(Y_j = m|Y_i = n, (i,j) \in \mathcal{E})$ is the conditional probability of a label given the neighborhood label. We analyze the conditional distribution in balanced binary classification. Let us define node-wise individual homophily $p_i$ and class homophily $c_k$ as:

$$p_i := \frac{|\{(i,j)|(i,j) \in \mathcal{E}, y_i = y_j\}|}{|\{(i,j)|(i,j) \in \mathcal{E}\}|},$$

$$c_k := \frac{|\{(i,j)|(i,j) \in \mathcal{E}, y_i = k, y_j = k\}|}{|\{(i,j)|(i,j) \in \mathcal{E}, y_i = k\}|} = \hat{P}(Y_j = k|Y_i = k, (i,j) \in \mathcal{E}).$$

With PosteL, the probability that the posterior label is the same as the ground truth label is given by:

$$P(Y_i = y_i|\{Y_j = y_j\}_{j \in \mathcal{N}(i)}) = \frac{c_{y_i}^{|\mathcal{N}(i)|p_i}(1 - c_{y_i})^{|\mathcal{N}(i)|(1-p_i)}}{c_{y_i}^{|\mathcal{N}(i)|p_i}(1 - c_{y_i})^{|\mathcal{N}(i)|(1-p_i)} + c_{y_i'}^{|\mathcal{N}(i)|(1-p_i)}(1 - c_{y_i'})^{|\mathcal{N}(i)|(p_i)}},$$

where $y_i$ is the ground truth label of node $i$ and $y_i'$ is the other label.

With homophilic graphs where $c_y > (1 - c_y), c_{y'} > (1 - c_{y'})$ and $p_i > (1 - p_i)$ generally, the posterior distribution of the ground truth label is higher than the negative label. With heterophilic graphs, where $c_y < (1 - c_y), c_{y'} < (1 - c_{y'})$ and $p_i < (1 - p_i)$ generally, the posterior distribution of the ground truth is also higher than the negative label.

A simple analysis shows that PosteL assigns a high probability to the ground truth label regardless of whether the graph is homophilic or heterophilic. The presence of the global label statistics $\hat{P}(Y_j = m|Y_i = n, (i, j) \in \mathcal{E})$ plays an important role in these cases.

**SALS (Wang et al., 2021)**  SALS interpolates the ground truth label with the neighborhood labels to generate a soft label. Hence, when the graph is heterophilic, the soft label is likely to be dominated by the negative labels. Figure 7 shows the example of SALS, where the soft label is dominated by the negative label.

**ALS (Zhou et al., 2023)**  The label smoothing of ALS consists of three processes: 1) ALS aggregates the neighborhood labels using the formula $\frac{1}{|\mathcal{N}(i)|} \sum_{j \in \mathcal{N}(i)} \mathbf{e}_j$, 2) the aggregated labels are interpolated with the ground truth label, and 3) the interpolated labels are transformed via a linear transform parameterized by learnable weight matrix $\mathbf{W}$ followed by softmax.

Analyzing the deterministic behavior of ALS (Zhou et al., 2023), as done previously for PosteL and SALS, is non-trivial due to the presence of a learnable component. However, there are explicit cases where ALS fails to distinguish nodes with different labels. When $p_i = 0$, ALS generates a soft label as softmax($[1, 1]\mathbf{W}$), regardless of the value of $y_i$. In such cases, ALS adds the same noise regardless of the characteristics of nodes, which is identical to uniform smoothing. Figure 7 illustrates an example of ALS, where it assigns the same label to all nodes.

**Global label distribution and imbalanced datasets**  Next, $\hat{P}(Y_i = m) := \frac{|\{u|y_u=m\}|}{|\mathcal{V}|}$, represents the proportion of label $m$ across all nodes. This enables PosteL to account for the overall label distribution in the graph. When a label has a low proportion, PosteL assigns it a lower probability. Reflecting the label distribution across the entire graph can be advantageous for graphs with imbalanced labels. For example, PosteL demonstrates consistent performance improvement on the imbalanced Computers dataset.

**Discussion**  Based on these differences, we argue that PosteL is significantly distinguished from other smoothing methods. This difference comes from the use of global statistics, $\hat{P}(Y_j = m|Y_i = n, (i, j) \in \mathcal{E})$, so we conjecture that it is the main factor behind PosteL's superior performance. This aligns with the results in Table 2 of our paper, which show that replacing global statistics with local statistics decreases performance.

## C.2 THE ADVANTAGE OF UTILIZING PSEUDO LABELING STRATEGY

Another distinction is the pseudo labeling strategy. SALS and ALS cannot work on sparse graphs, as there will be no labeled neighborhoods. Our pseudo labeling strategy enables smoothing using neighborhood label information even on sparse graphs. To the best of our knowledge, PosteL is the first label smoothing approach in node classification to address sparse label scenarios.

# D   DATASET STATISTICS

We provide detailed statistics and explanations about the dataset used for the experiments in Table 6 and the paragraphs below.

Table 6: Statistics of the dataset utilized in the experiments.

| Dataset | # nodes | # edges | # features | # classes |
|---|---|---|---|---|
| Cora | 2,708 | 5,278 | 1,433 | 7 |
| CiteSeer | 3,327 | 4,552 | 3,703 | 6 |
| PubMed | 19,717 | 44,324 | 500 | 3 |
| Computers | 13,752 | 245,861 | 767 | 10 |
| Photo | 7,650 | 119,081 | 745 | 8 |
| Chameleon | 2,277 | 31,396 | 2,325 | 5 |
| Actor | 7,600 | 30,019 | 932 | 5 |
| Squirrel | 5,201 | 198,423 | 2,089 | 5 |
| Texas | 183 | 287 | 1,703 | 5 |
| Cornell | 183 | 277 | 1,703 | 5 |

**Cora, CiteSeer, and PubMed**   Each node represents a paper, and an edge indicates a reference relationship between two papers. The task is to predict the research subjects of the papers.

**Computers and Photo**   Each node represents a product, and an edge indicates a high frequency of concurrent purchases of the two products. The task is to predict the product category.

**Chameleon and Squirrel**   Each node represents a Wikipedia page, and an edge indicates a link between two pages. The task is to predict the monthly traffic for each page. We use the classification version of the dataset, where labels are converted by dividing monthly traffic into five bins.

**Actor**   Each node represents an actor, and an edge indicates that two actors appear on the same Wikipedia page. The task is to predict the category of the actors.

**Texas and Cornell**   Each node represents a web page from the computer science department of a university, and an edge indicates a link between two pages. The task is to predict the category of each web page as one of the following: student, project, course, staff, or faculty.

# E   DETAILED EXPERIMENTAL SETUP

In this section, we provide the computer resources and search space for hyperparameters. Our experiments are executed on AMD EPYC 7513 32-core Processor and a single NVIDIA RTX A6000 GPU with 48GB of memory.

We use the same hyperparameter search space as He et al. (2021). Specifically, the learning rate is validated within $\{0.001, 0.002, 0.01, 0.05\}$, and weight decay within $\{0, 0.0005\}$. We set the number of layers for all models to two. The dropout ratio for the linear layers is fixed at 0.5. For the GCN (Kipf & Welling, 2016), the hidden layer dimension is set to 64. The GAT (Veličković et al., 2017) uses eight heads, each with a hidden dimension of eight. For the APPNP (Gasteiger et al., 2018), a two-layer MLP with a hidden dimension of 64 is used, the power iteration step is set to 10, and the teleport probability is chosen from $\{0.1, 0.2, 0.5, 0.9\}$. For the MLP, the hidden dimension is set to 64. For the ChebNet (Defferrard et al., 2016), the hidden dimension is set to 32, and two propagation steps are used. For the GPR-GNN (Chien et al., 2020), a two-layer MLP with a hidden dimension of 64 is used as the feature extractor neural network, and the random walk path length is set to 10. The PPR teleport probability is chosen from $\{0.1, 0.2, 0.5, 0.9\}$. For BernNet (He et al., 2021), a two-layer MLP with a hidden dimension of 64 is used as the feature extractor, and the polynomial approximation order is set to 10. The dropout ratio for the propagation layers in both GPR-GNN and BernNet is chosen from $\{0.0, 0.1, 0.2, 0.3, 0.4, 0.5, 0.6, 0.7, 0.8, 0.9\}$. We validate two

Table 7: Average iteration counts of iterative pseudo labeling for each backbone and dataset used to report Table 1.

|  | Cora | CiteSeer | PubMed | Computers | Photo | Chameleon | Actor | Squirrel | Texas | Cornell |
|---|---|---|---|---|---|---|---|---|---|---|
| GCN+PosteL | 2.5 | 2.2 | 1.5 | 1 | 0.9 | 0.9 | 1.1 | 0.7 | 1.8 | 2.5 |
| GAT+PosteL | 1.6 | 1.8 | 1 | 1.2 | 0.7 | 0.8 | 2 | 1.1 | 3.1 | 2.4 |
| APPNP+PosteL | 1.9 | 2 | 1.1 | 0.8 | 1.1 | 1 | 1.1 | 0.9 | 1.4 | 2.9 |
| MLP+PosteL | 1.7 | 2.2 | 0.4 | 0.7 | 0.7 | 0.1 | 0.8 | 0.6 | 0.9 | 2.4 |
| ChebNet+PosteL | 1.6 | 2.1 | 1.2 | 0.6 | 0.6 | 1 | 0.7 | 0.7 | 2 | 2 |
| GPR-GNN+PosteL | 0.8 | 1.1 | 0.8 | 0.5 | 1.3 | 1 | 0.3 | 0.7 | 1.1 | 1 |
| BernNet+PosteL | 1.5 | 1.8 | 0.9 | 0.8 | 1 | 1.5 | 1.5 | 0.5 | 1.2 | 2.1 |

hyperparameters for PosteL: posterior label ratio $\alpha \in \{0.1, 0.2, 0.3, 0.4, 0.5, 0.6, 0.7, 0.8, 0.9, 1.0\}$ and uniform noise ratio $\beta \in \{0, 0.1, 0.2, 0.3, 0.4, 0.5, 0.6, 0.7, 0.8, 0.9\}$.

# F  COMPLEXITY ANALYSIS

In this section, we provide a detailed analysis of the time complexity of Section 3.1. Specifically, we demonstrate the time complexity of obtaining the prior and likelihood distributions separately. Finally, we determine the time complexity of computing the posterior distribution using these distributions.

First, the prior distribution $\hat{P}(Y_i = m)$ can be obtained as follows:

$$\hat{P}(Y_i = m) = \frac{|\{u \mid y_u = k\}|}{|\mathcal{V}|} = \frac{\sum_{u \in \mathcal{V}} e_{um}}{|\mathcal{V}|}. \tag{18}$$

The time complexity of calculating Equation (18) is $O(|\mathcal{V}|)$, so the time complexity of calculating the prior distribution for $K$ classes is $O(|\mathcal{V}|K)$.

Next, calculating the empirical conditional $\hat{P}(Y_j = m | Y_i = n, (i, j) \in \mathcal{E})$ from Equation (2) can be performed as follows:

$$\hat{P}(Y_j = m | Y_i = n, (i, j) \in \mathcal{E}) \propto \sum_{u: u \in \mathcal{V}, y_u = n} \sum_{v \in \mathcal{N}(u)} e_{vm}. \tag{19}$$

The time complexity of calculating Equation (19) for all possible pairs of $m$ and $n$ is $O(\sum_{u \in \mathcal{V}} |\mathcal{N}(u)|K)$. Since $\sum_{u \in \mathcal{V}} \mathcal{N}(u) = 2|\mathcal{E}|$, the time complexity for calculating empirical conditional is $O(|\mathcal{E}|K)$.

The likelihood is approximated through the product of empirical conditional distributions, denoted as $P(\{Y_j = y_j\}_{j \in \mathcal{N}(i)} | Y_i = k) \approx \prod_{j \in \mathcal{N}(i)} \hat{P}(Y_j = y_j | Y_i = k, (i, j) \in \mathcal{E})$. Likelihood calculation for all training nodes operates in $O(\sum_{u \in \mathcal{V}} |\mathcal{N}(u)|K)$ time complexity. So the overall computational complexity for likelihood calculation is $O(|\mathcal{E}|K)$.

After obtaining the prior distribution and likelihood, the posterior distribution is obtained by Bayes' rule in Equation (1). Applying Bayes' rule for $|\mathcal{V}|$ nodes and $K$ classes can be done in $O(|\mathcal{V}|K)$. So the overall time complexity is $O((|\mathcal{E}| + |\mathcal{V}|)K)$. In most cases, $|\mathcal{V}| < |\mathcal{E}|$, so the time complexity of PosteL is $O(|\mathcal{E}|K)$.

In Section 3.2, iterative pseudo labeling is proposed, which involves iteratively refining the pseudo labels of validation and test nodes to calculate posterior labels. Since this process requires training the model from scratch for each iteration, the number of iterations can be a significant bottleneck in terms of runtime. Consequently, the iteration counts are evaluated to assess this aspect. The mean iteration counts for each backbone and dataset in Table 1 are summarized in Table 7. With an overall mean iteration count of 1.13, we argue that this level of additional time investment is justifiable for the sake of performance enhancement.

Table 8 shows the training time of PosteL and the other baselines. With IPL, PosteL requires more training time, being 1.3 times slower than ALS and 5.7 times slower than using GT labels. If this computational overhead is too heavy, we can use PosteL without IPL or IPL with one iteration as

an alternative. PosteL without IPL is 2 times faster than KD and ALS, and PosteL with IPL with one iteration is also faster than KD and ALS while not sacrificing the accuracy. We reported the accuracy of each variation in Table 3.

Table 8: Overall training time for each smoothing method. PosteL (w/o) refers to PosteL without IPL, and PosteL (1) refers to PosteL with one iteration of pseudo labeling.

|  | GCN | +LS | +KD | +SALS | +ALS | +PosteL | +PosteL (w/o) | +PosteL (1) |
|---|---|---|---|---|---|---|---|---|
| time (s) | 0.91 | 0.74 | 3.54 | 0.79 | 3.92 | 5.19 | 1.65 | 3.12 |

# G  ADDITIONAL EXPERIMENTS

**Training with extremely sparse labels**   We evaluate the performance of PosteL with extremely sparse labels. For all datasets, we randomly select 10 nodes per class as training nodes, resulting in $10K$ training nodes. In the case of the PubMed dataset, only 0.15% of nodes are used as training nodes. Table 9 shows the performance of each label smoothing method with extremely sparse labels on GCN. PosteL outperforms the baselines even in this extremely sparse setting, particularly on heterophilic datasets, demonstrating that the pseudo-labeling strategy effectively mitigates the issue of sparsity.

Table 9: The accuracy of the model trained with extremely sparse labels on GCN.

|  | Cora | CiteSeer | PubMed | Computers | Photo | Chameleon | Actor | Squirrel | Texas | Cornell |
|---|---|---|---|---|---|---|---|---|---|---|
| GCN | 76.75±0.63 | 66.28±0.96 | 76.74±1.28 | 75.25±2.61 | 89.34±1.20 | 40.01±1.74 | 21.06±1.84 | 25.92±1.91 | 64.19±2.86 | 58.67±3.52 |
| +LS | 77.05±0.72 | 65.88±1.18 | 76.88±1.29 | 76.73±2.65 | 89.10±1.08 | 40.48±1.78 | 21.37±1.37 | 26.40±1.76 | 63.71±2.57 | 59.52±2.95 |
| +SALS | **77.09±0.94** | 66.36±1.00 | 76.85±1.21 | 76.23±2.53 | 88.80±1.20 | 39.54±1.56 | 20.90±2.46 | 26.20±2.06 | 62.86±4.76 | 55.90±5.05 |
| +ALS | 76.71±0.55 | **66.46±1.00** | 77.09±1.33 | 75.92±2.54 | 89.69±0.96 | 40.62±1.96 | 22.32±1.72 | 25.64±1.99 | 65.14±2.67 | 58.57±3.33 |
| +PosteL | 77.00±0.80 | 66.21±1.01 | **77.35±1.28** | **77.88±2.25** | **89.78±0.87** | **43.24±1.22** | **25.76±0.69** | **27.89±1.36** | **66.10±2.76** | **63.14±2.10** |

**Scalability to large-scale graphs**   We measured the runtime of PosteL on the ogbn-products dataset (Hu et al., 2020), which contains 2,449,029 nodes and 61,859,140 edges, to validate the computational complexity on a large-scale graph. We measured the time excluding the training time for iterative pseudo labeling. Using PosteL, generating soft labels takes 52.57 seconds, while training for one epoch requires 19.11 seconds. These results indicate that PosteL can efficiently generate soft labels, even on large-scale graph structures.

Table 10: The accuracy of label smoothing methods on the ogbn-products dataset using GCN.

|  | GCN | +LS | +SALS | +ALS | +PosteL |
|---|---|---|---|---|---|
| ogbn-products | 80.62±0.68 | 80.99±0.50 | 81.12±0.13 | 80.46±0.38 | **81.20±0.68** |

Table 10 shows PosteL's performance on the ogbn-products dataset on GCN. While the performance improvement is not statistically significant, PosteL achieves the best performance compared to other smoothing methods.

# H    LEARNING CURVES ANALYSIS FOR ALL DATASETS

The learning curves for all datasets are provided in Figure 8 and Figure 9.

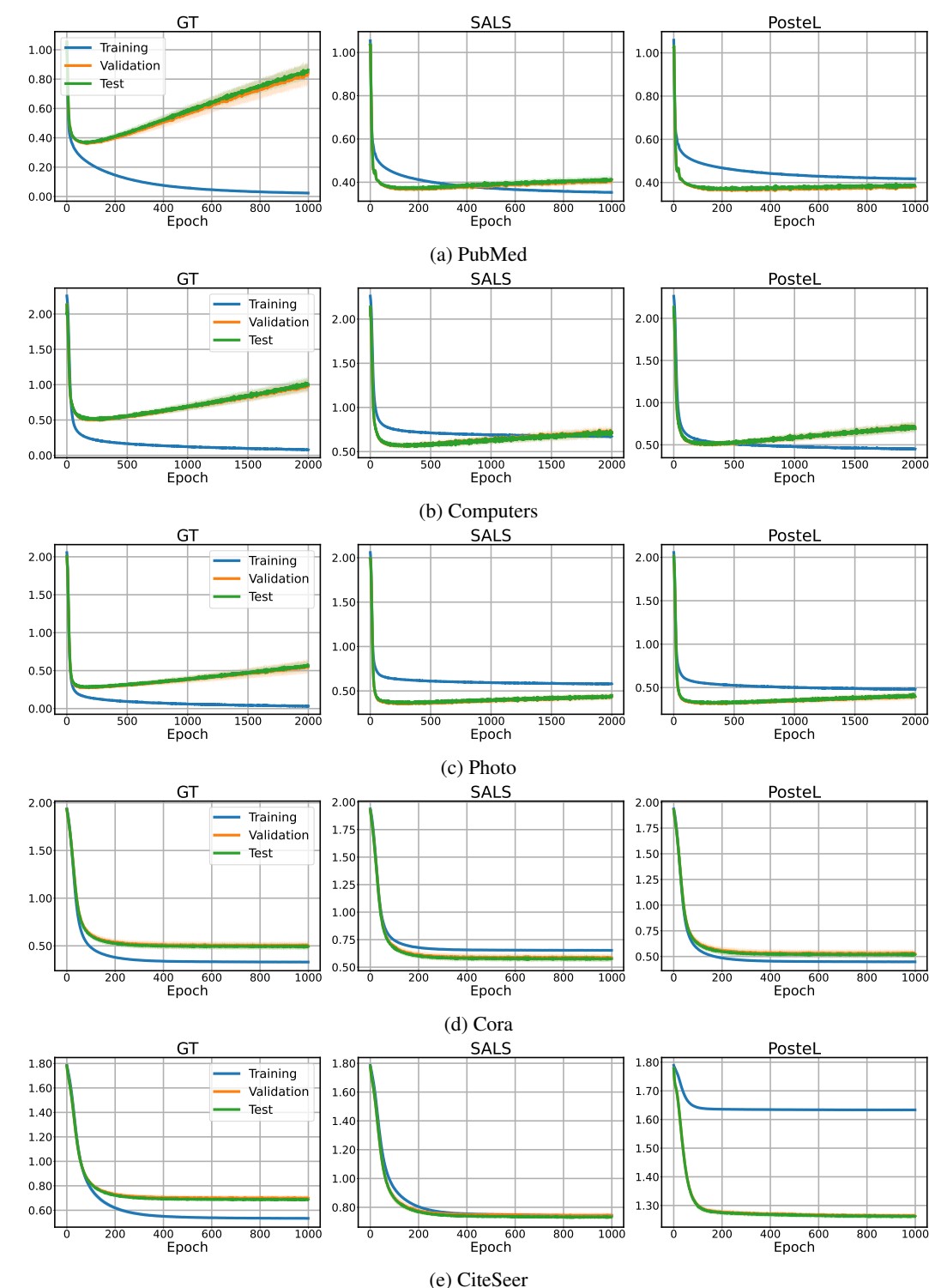

Figure 8: Loss curve of GCN trained on PosteL labels, SALS labels, and ground truth labels on homophilic datasets.

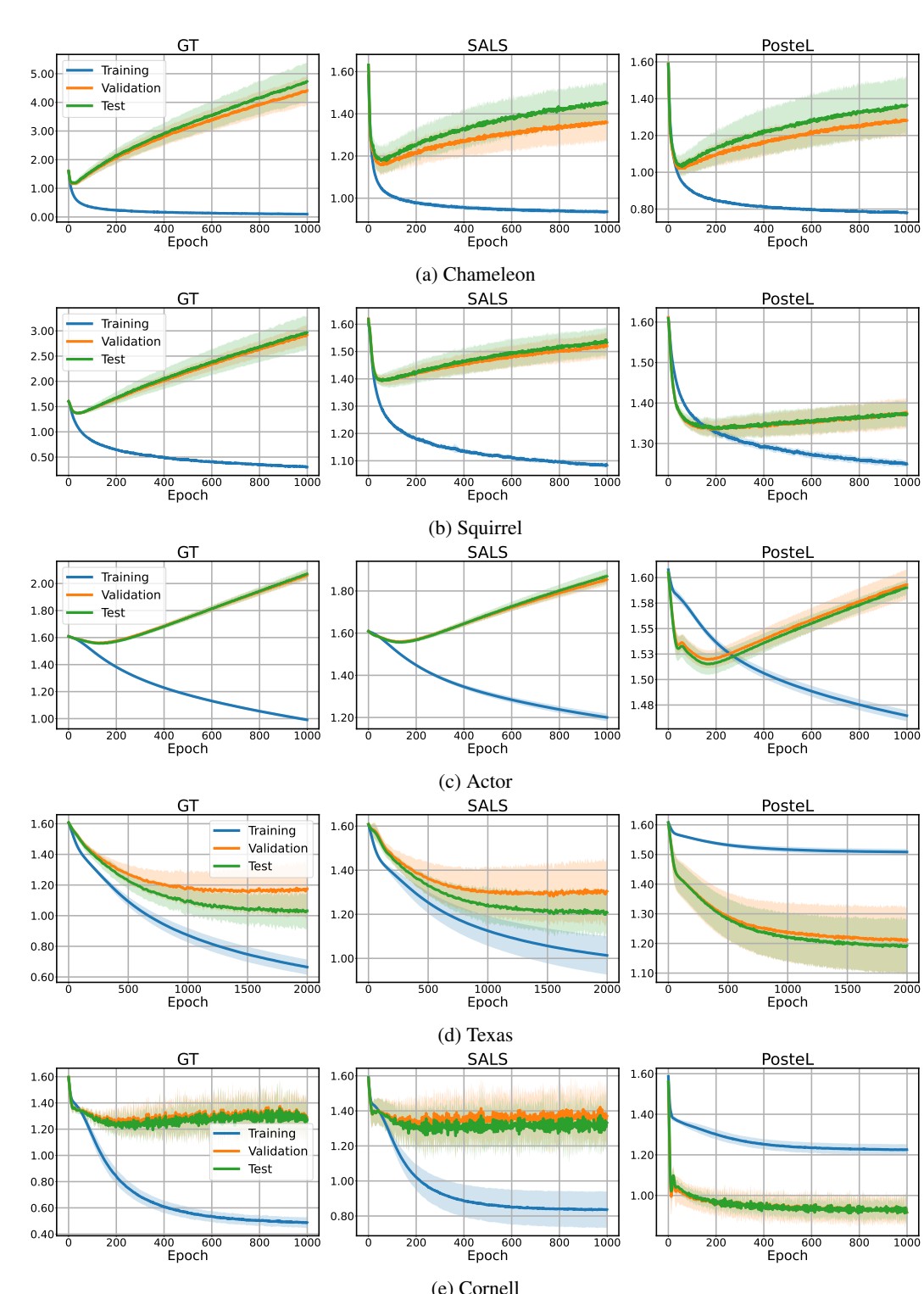

Figure 9: Loss curve of GCN trained on PosteL labels, SALS labels, and ground truth labels on heterophilic datasets.

## I EMPIRICAL CONDITIONAL DISTRIBUTION FOR ALL DATASETS

The empirical conditional distribution for all datasets is provided in Figure 10 and Figure 11.

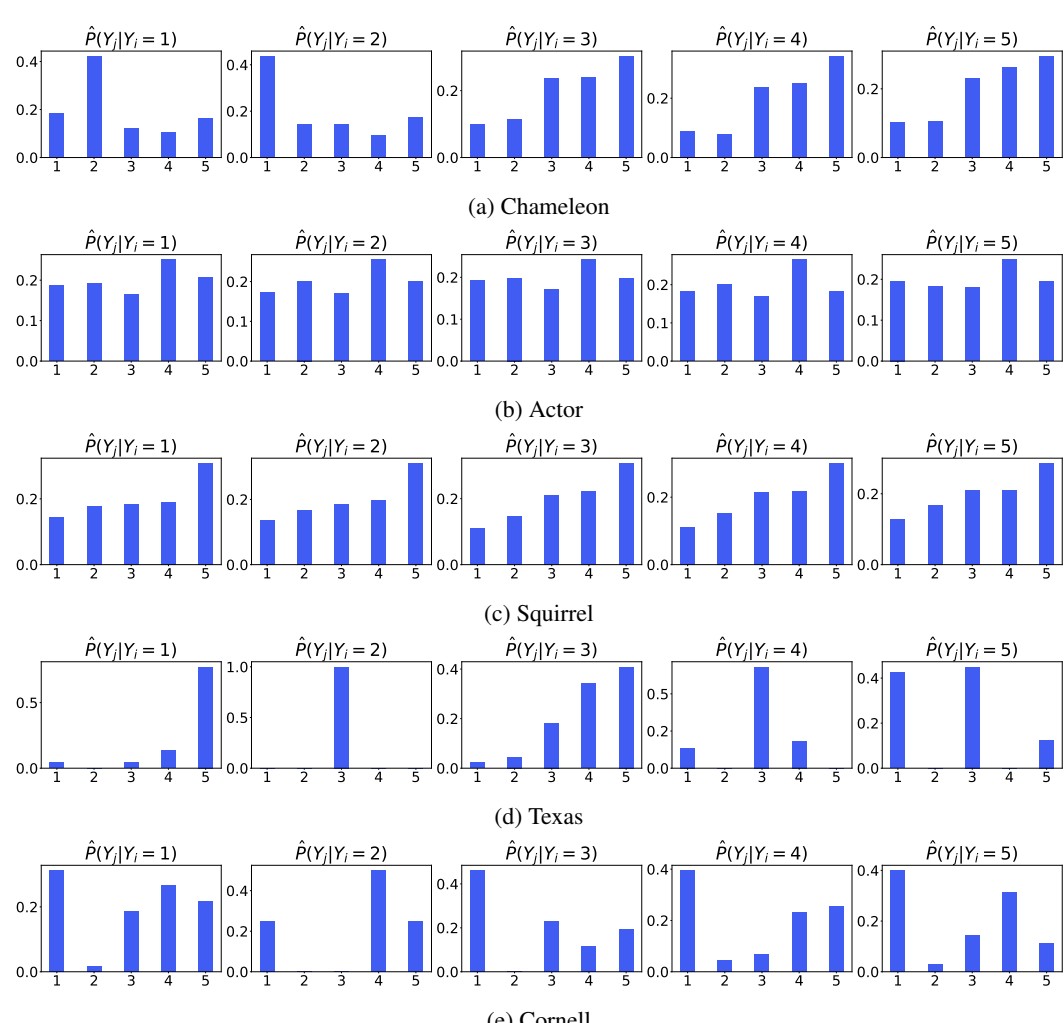

Figure 10: Empirical conditional distributions between two adjacent nodes on heterophilic graphs.

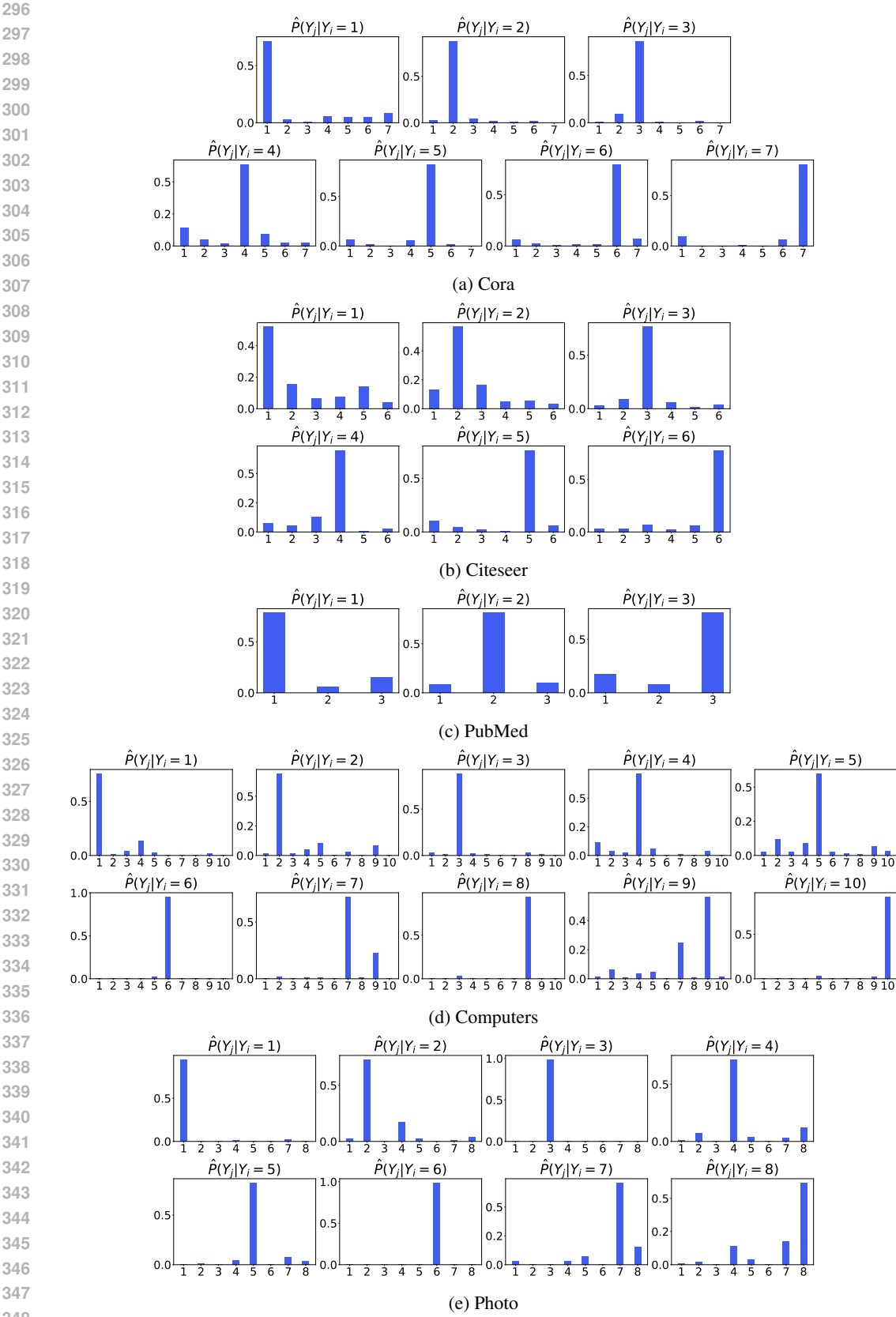

Figure 11: Empirical conditional distributions between two adjacent nodes on homophilic graphs.

