# OpenReview forum: "Posterior Label Smoothing for Node Classification"
_ICLR.cc/2025/Conference — Submitted to ICLR 2025_

### Official Review · Reviewer_Uap1 · 2024-10-22

**Soundness:** 2
**Presentation:** 4
**Contribution:** 2
**Rating:** 6
**Confidence:** 4

**Summary:**

The paper proposes a label-smoothing method called PosteL for the transductive node classification task. This method generates soft labels by integrating local neighborhood information and global label statistics, enhancing generalization and reducing overfitting. Specifically, it computes the posterior distribution of node labels by estimating conditional distributions through neighbour nodes and prior distributions through all labeled nodes, and then interpolates the posterior distribution with the ground truth label to obtain a soft label.Experiments on 10 node classification datasets with 7 baseline models demonstrate its effectiveness.

**Strengths:**

1. This paper introduces a simple yet effective label smoothing method for transductive node classification.

2. This paper investigates the performance of the proposed label smoothing method on different datasets and models, and analyzes the key factors for its success.

3. The authors present their ideas in a well-structured manner, making it easy for readers to follow the flow of the research. The figures in the paper are presented clearly and effectively support the text.

**Weaknesses:**

1. After claiming the effective handling of heterophilic graphs as a highlight, the description of the proposed method fails to emphasize its treatment of heterophilic graphs. For instance, it is not clear why the proposed method is suitable for handling heterophilic graphs and where the efficiency of this treatment is manifested.

2. The motivation for proposing the method and the effects achieved by the method itself seem to be contradictory. In Section 1, the authors state "...their performance on heterophilic graphs, where nodes tend to connect with others that are dissimilar or belong to different classes, still remains questionable." However, the proposed method estimates the posterior of one node based on label frequency of its neighbours, which leads to its prediction similar to the classes of its neighbours. In addition, the estimation of the prior mitigates the problem of class imbalance but does not address the stated problem of heterogeneity.

3. The idea in the method section of the article is similar to that of a naive Bayes approach, but some assumptions of conditional independence need to be clearly stated in Section 3.

4. In the case where the neighbors are not labeled and the neighbors of the neighbors are also not labeled, an iterative process of generating pseudo-labels needs to be reflected in this algorithm part.

**Questions:**

1. Please explain how the method proposed in the article addresses the stated problem of heterogeneity.

2. In the setting of this paper,  the connectivity between all nodes, including the test nodes, is assumed to be observed. Is this assumption a bit too strong? Please give some explanations.

---

> ### Author Response · Authors · 2024-11-21
>
> We sincerely appreciate the dedicated effort in evaluating our work and providing constructive feedback. We recognize that the explanation for why our method works on heterophilic graphs was insufficient, and we believe the response will help clarify this. A detailed explanation is provided below, and if any parts remain unclear or if further questions arise, we would be happy to address them.
>
> >**W1: After claiming the effective handling of heterophilic graphs as a highlight, the description of the proposed method fails to emphasize its treatment of heterophilic graphs. For instance, it is not clear why the proposed method is suitable for handling heterophilic graphs and where the efficiency of this treatment is manifested.**
>
> >**Q1: Please explain how the method proposed in the article addresses the stated problem of heterogeneity.**
>
> We provide an in-depth analysis of the underlying assumptions of PosteL and a case study comparing smoothing methods and PosteL in the general response at the top of the page. Please refer to the general response for details.
>
> Firstly, PosteL assumes that nodes with similar neighborhood label distributions should exhibit similar characteristics. As shown in the general response, this assumption holds for both homophilic and heterophilic graphs. We conjecture that this generality of the assumption contributes to PosteL's strong performance across both types of datasets.
>
> Next, PosteL assigns a high probability to the ground truth label for both homophilic and heterophilic graphs. This is in evident difference to SALS, which is dominated by negative labels on heterophilic graphs.
>
> Comparing PosteL to ALS is non-trivial due to the trainable component in ALS. However, we found that ALS fails to distinguish nodes with different labels when individual node homophily is zero. We also show that PosteL can distinguish the failure cases of ALS.
>
> This difference comes from the use of global statistics, $\hat{P}(Y_j=m|Y_i=n, (i,j) \in \mathcal{E})$, so we conjecture that it is another main factor behind PosteL's superior performance. This aligns with the results in Table 2 of our paper, which show that replacing global statistics with local statistics decreases performance.
>
> >**W2: The motivation for proposing the method and the effects achieved by the method itself seem to be contradictory. In Section 1, the authors state "...their performance on heterophilic graphs, where nodes tend to connect with others that are dissimilar or belong to different classes, still remains questionable." However, the proposed method estimates the posterior of one node based on label frequency of its neighbours, which leads to its prediction similar to the classes of its neighbours. In addition, the estimation of the prior mitigates the problem of class imbalance but does not address the stated problem of heterogeneity.**
>
> As analyzed in the general response and the [attached example](https://ibb.co/gVD9rBs), the PosteL label differs from the labels of neighboring nodes on heterophilic graphs. Figure 1 in the paper also illustrates this, where the soft label assigns a high probability to the green label, despite neighboring nodes being labeled red and blue. For these reasons, we respectfully disagree with this comment.
>
> >**W3: The idea in the method section of the article is similar to that of a naive Bayes approach, but some assumptions of conditional independence need to be clearly stated in Section 3.**
>
> Our method is based on the assumption that neighborhood labels are conditionally independent, given the label of the node to be relabeled. The approximation described in lines 153–156 stems from this assumption. We have clarified this point in the revised version (lines 151–153).
>
> >**W4: In the case where the neighbors are not labeled and the neighbors of the neighbors are also not labeled, an iterative process of generating pseudo-labels needs to be reflected in this algorithm part.**
>
> When there is no labeled neighborhood, the likelihood cannot be calculated. In such cases, we use the ground truth label instead of the posterior distribution. We have clarified this exception in the revised version of our paper, specifically in Algorithm 1 and Algorithm 2. To minimize the occurrence of such cases, we propose iterative pseudo labeling. This method assigns pseudo labels using the trained GNN, ensuring that every node has either a label or a pseudo label. As a result, exception cases arise only for nodes with no neighbors. For a detailed explanation, please refer to Algorithms 2 and 3 in Appendix B. Additionally, the absence of labels in neighbors of neighbors does not pose an issue for our method, as it does not rely on the labels of two-hop neighbors.

---

> ### Author Response · Authors · 2024-11-21
>
> >**Q2: In the setting of this paper, the connectivity between all nodes, including the test nodes, is assumed to be observed. Is this assumption a bit too strong? Please give some explanations.**
>
> This problem setup is known as the transductive setting. In a transductive setting, the labels of training nodes and all connectivity information, including that of test nodes, are provided. The objective of this setting is to predict the labels of test nodes using their connectivity information. This is a common setting in node classification papers [1, 2, 3]. A typical example is the citation graph, where the objective is to predict the category of a paper using the information from adjacent (cited) papers.
>
> [1] Kipf, Thomas N., and Max Welling. "Semi-supervised classification with graph convolutional networks." ICLR 2017.
> [2] Veličković, Petar, et al. "Graph attention networks." ICLR 2018.
> [3] Chien, Eli, et al. "Adaptive universal generalized pagerank graph neural network." ICLR 2021.

---

> > ### Comment · Reviewer_Uap1 · 2024-11-27
> >
> > The author has already solved most of my questions. For W1 and Q1, the direct reason why the proposed method could deal with heterophilic graphs instead of the experimental discovery or the statement of the employed assumption is supposed to be given, which will make the entire article more persuasive.

---

> ### Author Response · Authors · 2024-11-28
>
> We sincerely appreciate the response. We believe the provided feedback is very important and it will be thoroughly addressed in this response. It would be greatly appreciated if this could be reviewed with kind consideration. We have included this analysis in Appendix A of the revised version.
>
> ### **Theoretical analysis of the characteristics of PosteL with heterophilic graphs**
>
> In addition to the detailed analysis provided in the general response (at the top of the page), we provide additional two lemmas showing the characteristics of PosteL with heterophilic graphs.
>
> For simplicity, we focus on a binary classification problem with two classes: 0 and 1. To establish the lemma, we define the individual node homophily $p_i$ and class homophily $c_k$ as follows:
>
> \begin{equation}
> p_i:=\frac{|\\{(i,j)|(i,j)\in\mathcal{E},y_i=y_j\\}|}{|\\{(i,j)|(i,j)\in\mathcal{E}\\}|}, c_k:=\frac{|\\{(i,j)|(i,j)\in\mathcal{E},y_i=k,y_j=k\\}|}{|\\{(i,j)|(i,j)\in\mathcal{E},y_i=k\\}|}.
> \end{equation}
>
> Then the posterior label of the node being labeled to 0 is:
>
> \begin{equation}
> P(Y_i=0|\\{Y_j = y_j|j\in\mathcal{N}(i)\\})  = \frac{c_0^{|\mathcal{N}(i)|p_i}(1-c_0)^{|\mathcal{N}(i)|(1-p_i)}}{c_0^{|\mathcal{N}(i)|p_i}(1-c_0)^{|\mathcal{N}(i)|(1-p_i)}+c_1^{|\mathcal{N}(i)|(1-p_i)}(1-c_1)^{|\mathcal{N}(i)|p_i}}.
> \end{equation}
>
> **Lemma 1. In a heterophilic graph with $c_0, c_1 < 0.5$, if two nodes have the same degree $d$, and node $i$ is connected to more nodes labeled 1 than node $j$, then PosteL assigns a higher probability of node $i$ being labeled 0 compared to node $j$.**
>
> Since the number of adjacent nodes with a different label is larger for node $i$, we have $p_i<p_j$. The lemma can be expressed as follows:
>
> \begin{equation}
> \frac{c_0^{dp_i}(1-c_0)^{d(1-p_i)}}{c_0^{dp_i}(1-c_0)^{d(1-p_i)}+c_1^{d(1-p_i)}(1-c_1)^{dp_i}}>\frac{c_0^{dp_j}(1-c_0)^{d(1-p_j)}}{c_0^{dp_j}(1-c_0)^{d(1-p_j)}+c_1^{d(1-p_j)}(1-c_1)^{dp_j}}.
> \end{equation}
>
> Expanding the inequality:
>
> \begin{align}
> c_0^{d(p_i+p_j)}(1-c_0)^{d(2-p_i-p_j)}+c_0^{dp_i}(1-c_0)^{d(1-p_i)}c_1^{d(1-p_j)}(1-c_1)^{dp_j} >\\
> c_0^{d(p_i+p_j)}(1-c_0)^{d(2-p_i-p_j)}+c_0^{dp_j}(1-c_0)^{d(1-p_j)}c_1^{d(1-p_i)}(1-c_1)^{dp_i}.
> \end{align}
>
> Subtracting $c_0^{d(p_i + p_j)}(1 - c_0)^{d(2 - p_i - p_j)}$ from both sides:
> \begin{equation}
>     c_0^{dp_i}(1-c_0)^{d(1-p_i)}c_1^{d(1-p_j)}(1-c_1)^{dp_j}>c_0^{dp_j}(1-c_0)^{d(1-p_j)}c_1^{d(1-p_i)}(1-c_1)^{dp_i}.
> \end{equation}
>
> \begin{equation}
>     ((1-c_0)(1-c_1))^{d(p_j-p_i)}>(c_0c_1)^{d(p_j-p_i)}.
> \end{equation}
>
> Since $c_0<1-c_0$ and $c_1<1-c_1$ imply that $(1 - c_0)(1 - c_1)>c_0c_1$, and given $(1 - c_0)(1 - c_1)>c_0c_1$ and $p_j-p_i> 0$, the inequality holds.
>
> **Lemma 2. In a heterophilic graph with $c_0, c_1 < 0.5$, if two nodes are connected only to nodes labeled 1, and their respective degrees are $n$ and $m$ ($n > m$), then PosteL assigns a higher probability of being labeled 0 to the node with the higher degree.**
>
> When nodes are connected only to nodes labeled 1, PosteL assigns the posterior probability of the node being labeled 0 as follows:
> \begin{equation}
>     P(Y_i=0|\\{Y_j = 1|j\in\mathcal{N}(i)\\})=  \frac{(1-c_0)^{\text{deg}(i)}}{(1-c_0)^{\text{deg}(i)}+c_1^{\text{deg}(i)}},
> \end{equation}
>
> where $\text{deg}(i)$ represents the degree of node $i$. The lemma can be expressed as follows:
> \begin{equation}
>     \frac{(1-c_0)^n}{(1-c_0)^n+c_1^n} > \frac{(1-c_0)^m}{(1-c_0)^m+c_1^m}.
> \end{equation}
>
> Expanding the inequality:
>
> \begin{equation}
>     ((1-c_0)^m+c_1^m)(1-c_0)^n > ((1-c_0)^n+c_1^n)(1-c_0)^m.
> \end{equation}
>
> \begin{equation}
>     (1-c_0)^n(1-c_0)^m+c_1^m(1-c_0)^n > (1-c_0)^n(1-c_0)^m+c_1^n(1-c_0)^m.
> \end{equation}
>
> \begin{equation}
>     c_1^m(1-c_0)^n > c_1^n(1-c_0)^m.
> \end{equation}
>
> \begin{equation}
>     (1-c_0)^{n-m} > c_1^{n-m}.
> \end{equation}
>
> Since $c_0,c_1<0.5$, it follows that $1-c_0>c_1$. Thus, the inequality holds.
>
> As a corollary we can show the following two properties.
>
> 1. If two nodes have the same degree, the node connected to more nodes labeled 1 should have a higher probability of being labeled 0.
> 2. If two nodes are connected only to nodes labeled 1, the node with the higher degree should have a higher probability of being labeled 0.
>
> Although these lemmas and corollary may not reflect the real-world scenario, analyzing properties of a model is an important step towards understanding its performance.

---

> ### Author Response · Authors · 2024-11-28
>
> ### **The comparison between PosteL and the other smoothing methods**
>
> (This section is copied from the general response in case, it is missed. The analysis is not just showing the assumptions and empirical results but contains key characteristics of PosteL to compare with the existing methods.)
>
> **1. The effect of utilizing global statistics**
> PosteL leverages global statistics, specifically $\hat{P}(Y_j=m|Y_i=n, (i,j)\in\mathcal{E})$ and $\hat{P}(Y_i=m)$, to generate soft labels. In contrast, SALS [1] only utilizes information from 1-hop neighbors. The use of global statistics in ALS [2] is questionable due to the presence of learnable component in their soft label. We provide an example of the soft labels on a binary node classification task with a bipartite graph at this [**(anonymized link)**](https://ibb.co/gVD9rBs). The toy example highlights the key differences between existing methods and ours, which will be elaborated further below.
>
> **1-1. Conditional probability $\hat{P}(Y_j=m|Y_i=n)$ and its impact**
> $\hat{P}(Y_j=m|Y_i=n, (i,j)\in\mathcal{E})$ is the conditional probability of a label given the neighborhood label. We analyze the effect of the conditional distribution in balanced binary classification. With PosteL, the probability that the posterior label is the same as the ground truth label is given by:
>
> \begin{equation}
> P(Y_i=y_i|\\{Y_j = y_j|j\in\mathcal{N}(i)\\})  = \frac{c_{y_i}^{|\mathcal{N}(i)|p_i}(1-c_{y_i})^{|\mathcal{N}(i)|(1-p_i)}}{c_{y_i}^{|\mathcal{N}(i)|p_i}(1-c_{y_i})^{|\mathcal{N}(i)|(1-p_i)}+c_{y_i'}^{|\mathcal{N}(i)|(1-p_i)}(1-c_{y_i'})^{|\mathcal{N}(i)|p_i}},
> \end{equation}
>
> where $y_i$ is the ground truth label of node $i$ and $y_i'$ is the other label.
>
> With homophilic graphs where $c_y>(1-c_y), c_{y'}>(1-c_{y'})$ and $p_i > (1-p_i)$ generally, the posterior distribution of the ground truth label is higher than the negative label. With heterophilic graphs, where $c_y<(1-c_y), c_{y'}<(1-c_{y'})$ and $p_i < (1-p_i)$ generally, the posterior distribution of the ground truth is also higher than the negative label.
>
> A simple analysis shows that PosteL assigns a high probability to the ground truth label regardless of whether the graph is homophilic or heterophilic. The presence of the global label statistics $\hat{P}(Y_j=m|Y_i=n, (i,j)\in\mathcal{E})$ plays an important role in these cases, as they provide information about the overall homophilicity of the graph.
>
> **SALS [1]**
> SALS interpolates the ground truth label with the neighborhood labels to generate a soft label. Hence, when the graph is heterophilic, the soft label is likely to be dominated by the negative labels. The figure in the [anonymized link](https://ibb.co/gVD9rBs) shows the example of SALS, where the soft label is dominated by the negative label.
>
> **ALS [2]**
> The label smoothing of ALS consists of three processes: 1) ALS aggregates the neighborhood labels using the formula $\frac{1}{|\mathcal{N}(i)|}\sum_{j\in\mathcal{N}(i)}\mathbf{e}_j$, 2) the aggregated labels are interpolated with the ground truth label, and 3) the interpolated labels are transformed via a linear transform parameterized by learnable weight matrix $\mathbf{W}$ followed by softmax. (Note that our analysis is based on the published version of ALS [2] and not arxiv version - the methods are slightly different between these two versions.)
>
> Analyzing the deterministic behavior of ALS [2], as done previously for PosteL and SALS, is non-trivial due to the presence of a learnable component. However, there are explicit cases where ALS fails to distinguish nodes with different labels. When $p_i=0$, ALS generates a soft label as $\text{softmax}([1,1]\mathbf{W})$, regardless of the value of $y_i$. In such cases, ALS adds the same noise regardless of the characteristics of nodes, which is identical to uniform smoothing. The attached figure illustrates a example of ALS, where it assigns the same label to all nodes.
>
> **Discussion**
> Based on these differences, we argue that PosteL is significantly distinguished from other smoothing methods. This difference comes from the use of global statistics, $\hat{P}(Y_j=m|Y_i=n, (i,j) \in \mathcal{E})$, so we conjecture that it is the main factor behind PosteL's superior performance. This aligns with the results in Table 2 of our paper, which show that replacing global statistics with local statistics decreases performance.
>
> [1] Wang, Yiwei, et al. "Structure-aware label smoothing for graph neural networks." arXiv preprint 2021.
> [2] Zhou, Kaixiong, et al. "Adaptive label smoothing to regularize large-scale graph training." SDM 2023.

---

> > ### Comment · Reviewer_Uap1 · 2024-11-29
> >
> > Thank you for the comprehensive response to my review. Your explanation has successfully addressed my concerns, and as a result, I am pleased to increase my review score.

---

> > > ### Author Response · Authors · 2024-11-29
> > >
> > > Thank you for taking the time to update your review and share your kind words. I truly appreciate your thoughtful feedback and the opportunity to address your concerns. It’s rewarding to know that our response was helpful. Thank you once again!

---

### Official Review · Reviewer_vWwq · 2024-10-28

**Soundness:** 3
**Presentation:** 3
**Contribution:** 2
**Rating:** 8
**Confidence:** 3

**Summary:**

The manuscript proposes PosteL, a label smoothing method utilizing posterior distribution for node classification in graph-structured data. It is basically a preprocessing method for GNNs, generating soft labels based on neighborhood context and global label statistics before the training phase.

**Strengths:**

1. The manuscript is well-structured and easy to comprehend. For example, Fig. 1 is clear and intuitive.
2. The method is simple yet effective. PosteL can be combined seamlessly with existing methods.
3. The manuscript presents a wealth of experimental results that highlight the potential of PosteL for performance enhancement.

**Weaknesses:**

The significant drawback I perceive in this manuscript is that while it explores the usefulness of smoothing in the graph domain, the underlying principles that contribute to its effectiveness are not sufficiently clarified. Providing a theoretical analysis of its effectiveness would significantly strengthen the authors' claims.

**Questions:**

1. According to Section 2.2, PosteL differs from existing works focused on smoothing in the graph domain in terms of assumptions. However, is there any analysis that demonstrates how these assumptions influence performance improvement? Are there specific examples that support the manuscript's assumptions? Providing this would be crucial for substantiating the novelty of the work.
2. According to the appendix, the datasets are all structured data. Can PosteL be applied to a broader range of research in the graph domain?

---

> ### Author Response · Authors · 2024-11-21
>
> We sincerely appreciate the dedicated effort in the review process and the recognition of the strengths of our work. We are also grateful for the constructive feedback. We acknowledge the insufficient explanation of PosteL's effectiveness and have addressed this through our response. If any parts remain unclear or if further questions arise, we would be happy to address them.
>
> >**W1: The significant drawback I perceive in this manuscript is that while it explores the usefulness of smoothing in the graph domain, the underlying principles that contribute to its effectiveness are not sufficiently clarified. Providing a theoretical analysis of its effectiveness would significantly strengthen the authors' claims.**
>
> We provide an in-depth analysis of the underlying assumptions of PosteL and a case study comparing smoothing methods and PosteL in the general response at the top of the page. Please refer to the general response for details.
>
> Firstly, PosteL assumes that nodes with similar neighborhood label distributions should exhibit similar characteristics. As shown in the general response, this assumption holds for both homophilic and heterophilic graphs. We conjecture that this generality of the assumption contributes to PosteL's strong performance across both types of datasets.
>
> Next, PosteL assigns a high probability to the ground truth label for both homophilic and heterophilic graphs. This is in evident difference to SALS, which is dominated by negative labels on heterophilic graphs.
>
> Comparing PosteL to ALS is non-trivial due to the trainable component in ALS. However, we found that ALS fails to distinguish nodes with different labels when individual node homophily is zero. We also show that PosteL can distinguish the failure cases of ALS.
>
> This difference comes from the use of global statistics, $\hat{P}(Y_j=m|Y_i=n, (i,j) \in \mathcal{E})$, so we conjecture that it is another main factor behind PosteL's superior performance. This aligns with the results in Table 2 of our paper, which show that replacing global statistics with local statistics decreases performance.
>
> >**Q1: According to Section 2.2, PosteL differs from existing works focused on smoothing in the graph domain in terms of assumptions. However, is there any analysis that demonstrates how these assumptions influence performance improvement? Are there specific examples that support the manuscript's assumptions? Providing this would be crucial for substantiating the novelty of the work.**
>
> We address this question in the general response. Please refer to the section **In-depth analysis of the underlying assumptions of PosteL** in the general response.
>
> >**Q2: According to the appendix, the datasets are all structured data. Can PosteL be applied to a broader range of research in the graph domain?**
>
> The only requirements for applying PosteL are the node labels and adjacency information, which are typically provided in node classification tasks. Therefore, PosteL can be applied to a wider range of node classification researchs in the graph domain.

---

### Official Review · Reviewer_5wQb · 2024-11-01

**Soundness:** 3
**Presentation:** 3
**Contribution:** 3
**Rating:** 5
**Confidence:** 1

**Summary:**

This paper proposes a posterior label smoothing method (PosteL) that enhances node classification accuracy by incorporating both local neighborhood information and global label statistics. The approach demonstrates performance improvements on both homogeneous and heterogeneous graphs. Experimental results show that this method prevents overfitting and exhibits generalization capabilities across multiple datasets and baseline models.

**Strengths:**

1. This paper introduces a new idea in label smoothing by using posterior distribution calculations for label smoothing. Unlike traditional uniform noise smoothing methods, the authors propose combining local neighborhood information with global label distribution, allowing labels to reflect the local graph structure of nodes.
2. The method was tested on both standard homogeneous graphs and heterogeneous graphs, highlighting its general applicability. Experiments show that PosteL label smoothing can prevents overfitting and enhances model generalization without increasing model complexity.

**Weaknesses:**

1. While the posterior label smoothing method achieves notable improvements on graph-structured data, it relies on the neighborhood label distribution of nodes, which may lead to unstable posterior distributions in sparse graphs or scenarios with very few labels, resulting in reduced label quality. Although the authors propose a strategy of re-estimating the posterior with pseudo labels, limitations remain, especially on heterogeneous graphs with sparse labels.
2. Although the paper mentions that the computational complexity is linearly related to the number of edges and categories, it lacks sufficient discussion on efficient computation and optimization for large-scale graph structures, especially heterogeneous graphs. In real-world applications, such as large-scale social networks or e-commerce recommendation systems, this method may encounter efficiency bottlenecks.
3. Although the paper compares various existing label smoothing methods, it lacks adequate comparison with the latest graph classification models, particularly on large-scale heterogeneous graphs. This limitation suggests that PosteL’s effectiveness on more complex or dynamic graph tasks may require further validation.

**Questions:**

1 The author should clarify how different neighborhood configurations impact performance in heterophilic vs. homophilic settings.
2 How does the method perform with highly sparse labels, particularly below 10% labeled data? Are there specific mitigations for sparsity?
3  The aushor should provide details on the computational cost of PosteL relative to other smoothing methods, especially in terms of training duration across datasets?

---

> ### Author Response · Authors · 2024-11-21
>
> We sincerely appreciate the extensive effort in evaluating our paper and providing constructive feedback. We have carefully considered all comments and incorporated additional experiments in the revised version. If any parts remain unclear or if further questions arise, we would be happy to address them.
>
> Before addressing the feedback, we would like to clarify the distinction between the terms heterophilic and heterogeneous. Since our paper focuses on heterophilic graphs, we assume that the use of "heterogeneous" in the feedback refers to "heterophilic." If this assumption is incorrect, please let us know, and we will adjust our response accordingly.
>
> >**W1: While the posterior label smoothing method achieves notable improvements on graph-structured data, it relies on the neighborhood label distribution of nodes, which may lead to unstable posterior distributions in sparse graphs or scenarios with very few labels, resulting in reduced label quality. Although the authors propose a strategy of re-estimating the posterior with pseudo labels, limitations remain, especially on heterogeneous graphs with sparse labels.**
>
> >**Q2: How does the method perform with highly sparse labels, particularly below 10% labeled data? Are there specific mitigations for sparsity?**
>
> We evaluate the performance of PosteL with extremely sparse labels. For all datasets, we randomly select 10 nodes per class as training nodes, resulting in $10K$ training nodes. In the PubMed dataset, only 0.15% of nodes are used as training nodes. PosteL outperforms the baselines even in this extremely sparse setting, particularly on heterophilic datasets, demonstrating that the pseudo labeling strategy effectively mitigates the issue of sparsity. We have included this experiment in Appendix G of the revised version.
>
> ||Chameleon|Actor|Squirrel|Texas|Cornell|Cora|CiteSeer|PubMed|Computers|Photo|
> |-|-|-|-|-|-|-|-|-|-|-|
> |GCN        |40.0±1.7|21.1±1.8|25.9±1.9|64.2±2.9|58.7±3.5|76.8±0.6|66.3±1.0|76.7±1.3|75.3±2.6|89.3±1.2|
> |+LS        |40.5±1.8|21.4±1.4|26.4±1.8|63.7±2.6|59.5±3.0|**77.1±0.7**|65.9±1.2|76.9±1.3|76.7±2.7|89.1±1.1|
> |+SALS      |39.5±1.6|20.9±2.5|26.2±2.1|62.9±4.8|55.9±5.1|**77.1±0.9**|66.4±1.0|76.9±1.2|76.2±2.5|88.8±1.2|
> |+ALS       |40.6±2.0|22.3±1.7|25.6±2.0|65.1±2.7|58.6±3.3|76.7±0.6|**66.5±1.0**|77.1±1.3|75.9±2.5|89.7±1.0|
> |+PosteL    |**43.2±1.2**|**25.8±0.7**|**27.9±1.4**|**66.1±2.8**|**63.1±2.1**|77.0±0.8|66.2±1.0|**77.4±1.3**|**77.9±2.3**|**89.8±0.9**|
>
>
> >**W2: Although the paper mentions that the computational complexity is linearly related to the number of edges and categories, it lacks sufficient discussion on efficient computation and optimization for large-scale graph structures, especially heterogeneous graphs. In real-world applications, such as large-scale social networks or e-commerce recommendation systems, this method may encounter efficiency bottlenecks.**
>
> We measured the runtime of processing soft labels (Algorithm 1) on the ogbn-products dataset [1], which contains 2,449,029 nodes and 61,859,140 edges, to validate the computational complexity on a large-scale graph. Using our method, generating soft labels takes 52.57 seconds. Given that training for one epoch requires 19.11 seconds, this is not a significant overhead. These results demonstrate that PosteL can efficiently generate soft labels, even on large-scale graph structures. We have included this analysis in Appendix G of the revised version.

---

> ### Author Response · Authors · 2024-11-21
>
> >**W3: Although the paper compares various existing label smoothing methods, it lacks adequate comparison with the latest graph classification models, particularly on large-scale heterogeneous graphs. This limitation suggests that PosteL’s effectiveness on more complex or dynamic graph tasks may require further validation.**
>
> We validate the effectiveness of PosteL on the latest graph classification model, OrderedGNN [2]. PosteL achieves the best performance on nine out of ten datasets and significantly outperforms on the CiteSeer and Squirrel datasets. We have included this analysis in Table 1 of the revised version.
>
>
> |               |Cora|CiteSeer|PubMed|Computers|Photo|Chameleon|Actor|Squirrel|Texas|Cornell|
> |-|-|-|-|-|-|-|-|-|-|-|
> |**OGNN**      |88.6±1.1|80.1±0.9|88.7±0.6|89.7±0.5|94.8±0.4|58.3±1.3|39.7±1.2|38.7±1.1|**90.2±2.6**|90.3±2.5|
> |+LS       |88.5±0.9|80.2±0.8|88.2±0.3|89.6±0.5|94.5±0.5|58.9±1.6|40.0±0.7|40.1±0.8|88.2±3.6|91.2±1.3|
> |+SALS     |88.4±1.0|80.9±0.7|88.1±0.6|88.9±0.5|93.9±0.4|59.3±1.3|39.5±0.4|40.9±0.9|77.7±4.8|84.8±4.1|
> |+ALS      |88.0±0.7|80.6±0.6|88.7±0.6|89.8±0.5|94.8±0.4|59.4±1.2|40.3±0.8|40.4±1.1|90.0±2.6|89.8±3.0|
> |+PosteL  |**89.0±1.2**|**82.5±0.6**|**88.9±0.6**|**90.1±0.3**|**95.0±0.3**|**60.2±1.2**|**40.0±1.0**|**43.7±0.9**|87.7±5.3|**92.0±1.2**|
>
> We validate PosteL's performance on the large-scale ogbn-products dataset [1] on GCN. While the performance improvement is not statistically significant, PosteL achieves the best performance compared to other smoothing methods. We have included this experiment in Appendix G of the revised version.
>
> ||GT|LS|SALS|ALS|PosteL|
> |-|-|-|-|-|-|
> |ogbn-products|80.6±0.7|81.0±0.5|81.1±0.1|80.5±0.4|**81.2±0.7**|
>
> >**Q1: The author should clarify how different neighborhood configurations impact performance in heterophilic vs. homophilic settings.**
>
> We provide an in-depth analysis of the underlying assumptions of PosteL and a case study comparing smoothing methods and PosteL in the general response at the top of the page. Please refer to the general response for details.
>
> SALS demonstrates different neighborhood configurations could significantly impact performance. In homophilic graphs, the soft labels from SALS are dominated by positive labels. However, in heterophilic graphs, the soft labels from SALS are dominated by negative labels.
>
> Table 1 of our paper shows the differing effects of SALS on homophilic and heterophilic graphs. On homophilic graphs, SALS consistently performs better than the ground truth label. However, SALS shows a significant decrease in performance on the Texas dataset, which is the most heterophilic dataset in our experiments.
>
> In contrast, PosteL is not significantly affected by whether the graph is homophilic or heterophilic, as it consistently assigns a high probability to the ground truth label for both types of graphs.
>
> In addition, PosteL has the underlying assumption that nodes with similar neighborhood label distributions should exhibit similar characteristics. As shown in the general response, this assumption holds for both homophilic and heterophilic graphs. We conjecture that this generality of the assumption contributes to PosteL's strong performance across both types of datasets.
>
> >**Q3: The author should provide details on the computational cost of PosteL relative to other smoothing methods, especially in terms of training duration across datasets?**
>
> We estimated training time of PosteL and the other baselines and presented in the following table. With IPL, PosteL requires more training time, being 1.3 times slower than ALS and 5.7 times slower than using GT labels. If this computational overhead is too heavy, we can use PosteL without IPL or IPL with one iteration as an alternative. PosteL without IPL is 2 times faster than KD and ALS, and PosteL with IPL with one iteration is also faster than KD and ALS while not sacrificing the accuracy. We reported the accuracy of each variation in Table 3 of our paper. We have included this experiment in Appendix F of the revised version.
>
> ||Vanilla|LS|KD|SALS|ALS|PosteL|PosteL w/o IPL|PosteL with one iteration IPL|
> |-|-|-|-|-|-|-|-|-|
> |Time(s)|0.91|0.74|3.54|0.79|3.92|5.19|1.65|3.12|
>
>
> [1] Hu, Weihua, et al. "Open graph benchmark: Datasets for machine learning on graphs." NeurIPS 2020.
> [2] Song, Yunchong, et al. "Ordered gnn: Ordering message passing to deal with heterophily and over-smoothing." ICLR 2023.

---

> > ### Comment · Reviewer_5wQb · 2024-11-27
> >
> > Thank you for the detailed response. While the authors address several points raised in the review, there are still notable gaps and concerns. While the experiments on sparse labels are appreciated, the claim that PosteL performs well in such scenarios remains underexplored. The added results show performance improvements, but they lack detailed insights into how pseudo-labeling mitigates sparsity compared to other methods. This explanation is critical for understanding the robustness of PosteL under extreme conditions. The runtime analysis provides useful metrics but does not convincingly address concerns about scalability in real-world applications with heterogeneous or large-scale graphs. While generating soft labels may not add significant overhead, the overall computational cost of the framework, especially relative to simpler baselines, requires deeper exploration beyond one dataset. In summary, while the revisions address some concerns, the responses lack sufficient depth and clarity in key areas, leaving important questions about robustness and scalability.

---

> > > ### Author Response · Authors · 2024-11-29
> > >
> > > We sincerely appreciate the response. In this reply, we provide an in-depth analysis of sparse settings and further evaluate the runtime on multiple large-scale graphs. It would be greatly appreciated if this could be reviewed with kind consideration.
> > >
> > > > While the experiments on sparse labels are appreciated, the claim that PosteL performs well in such scenarios remains underexplored. The added results show performance improvements, but they lack detailed insights into how pseudo-labeling mitigates sparsity compared to other methods. This explanation is critical for understanding the robustness of PosteL under extreme conditions.
> > >
> > > The reason we conduct the experiment under the sparse setting is that PosteL empirically collects global statistics, but in sparse settings, this empirical distribution may be incorrect, resulting in low-quality soft labels. To provide insight into how PosteL mitigates the challenges of sparse settings, we analyze the empirical label distribution on the Cornell dataset, where iterative pseudo labeling demonstrates the largest improvements. The following table presents the performance with and without iterative pseudo labeling:
> > >
> > > ||Cornell (60% training)|Cornell (Extreme sparse)|
> > > |-|-|-|
> > > |w/o IPL|72.0±5.3|58.3±2.3|
> > > |w/ IPL|**80.3±1.8**|**63.1±2.1**|
> > >
> > > The [anonymous link](https://ibb.co/Ld6fJWB) shows the label distribution of training nodes (left), training nodes with pseudo labeled nodes (middle), and ground truth labels (right) on the Cornell dataset. The upper figures represent the distribution under the 60% training setting, while the lower figures show the distribution under the extremely sparse setting.
> > >
> > > We observe that the empirical label distribution from training nodes differs significantly from the ground truth label distribution, and pseudo labeling helps mitigate this difference. In the 60% training setting, the pseudo label distribution aligns well with the ground truth. Even in the extremely sparse setting, the pseudo label distribution appears similar to the ground truth, except for label 4.
> > >
> > > However, since pseudo labeling relies on the trained GNNs, its mitigating effect in the extremely sparse setting is less pronounced compared to the 60% training setting. Consequently, the performance gap is also larger in the 60% training setting.
> > >
> > > These results demonstrate the ability of pseudo labeling to address challenges in sparse settings, even in extreme cases. We hope this provides intuition for why PosteL performs well, even under sparse conditions.
> > >
> > > > The runtime analysis provides useful metrics but does not convincingly address concerns about scalability in real-world applications with heterogeneous or large-scale graphs. While generating soft labels may not add significant overhead, the overall computational cost of the framework, especially relative to simpler baselines, requires deeper exploration beyond one dataset.
> > >
> > > We provide the runtime of PosteL (Algorithm 1) on other large-scale graphs. The datasets are sourced from [1]. We further optimized the code to reduce runtime, so the runtime on ogbn-product differs from the previous response. The anonymous GitHub repository has been updated, and the updated version of PosteL can be found in lines 180–193 of relabel.py. Even on large-scale graphs, PosteL can process the soft label in under one minute.
> > >
> > > ||genius|pokec|ogbn-products|snap-patents|
> > > |-|-|-|-|-|
> > > |# nodes|421,961|1,632,803|2,449,029|2,923,922|
> > > |# edges|984,979|30,622,564|61,859,140|13,975,788|
> > > |Processing time|3.43s|13.58s|5.65s|25.05s|
> > >
> > > Indeed, PosteL requires more time than simpler baselines, such as uniform label smoothing, but a runtime increase of under one minute is acceptable for the sake of performance improvement. However, we acknowledge that iterative pseudo labeling could consume too much time on large-scale graphs. In such cases, avoiding iterative pseudo labeling might be a better choice. As shown in Table 3 of our paper, PosteL without iterative pseudo labeling still outperforms other baselines.
> > >
> > > [1] Lim, Derek, et al. "Large scale learning on non-homophilous graphs: New benchmarks and strong simple methods." NeurIPS 2021.

---

### Official Review · Reviewer_cvti · 2024-11-04

**Soundness:** 3
**Presentation:** 2
**Contribution:** 2
**Rating:** 3
**Confidence:** 4

**Summary:**

The paper introduces a method called Posterior Label Smoothing (PosteL) designed to improve node classification tasks in graph-structured data. By integrating both local neighborhood information and global label statistics, PosteL generates soft labels that enhance the generalization capabilities of models and mitigate overfitting. The method is applied to various datasets and models, showing that it consistently outperforms traditional label smoothing and other baseline methods.

**Strengths:**

1. The paper is well-written and the content is presented clearly, making it easy to follow.

2. The experiments are comprehensive and include a variety of different settings.

3. The proposed method is probabilistically driven, offering a more rational approach compared to existing heuristics.

**Weaknesses:**

1. The contribution of this paper is somewhat incremental - the idea of using neighborhood information to perform label smoothing has already been extensively studied [1, 2].

2. "Under the assumption that the neighborhood labels are conditionally independent given the label of the node to be relabeled...", I fail to comprehend this statement, hope authors can further explain this. But in any case, I think the assumption of two adjacent nodes are conditioally independent is too strong.

3. While the proposed method shows superior emprical performance, this paper fail to provide an in-depth explaination on why label-smoothing performs so well in graph-structured data - does it stems from the same reason as i.i.d. data? What makes the proposed method superior than other label smoothing methods on graph?

4. The improvements over the most competitive baselines are marginal - for most cases the differences are within the standard deviation.

[1] Adaptive Label Smoothing To Regularize Large-Scale Graph Training, Zhou et al., 2021.

[2] Structure-Aware Label Smoothing for Graph Neural Networks, Wang et al., 2021.

**Questions:**

please refer to the weakness

---

> ### Author Response · Authors · 2024-11-21
>
> We sincerely appreciate the effort in evaluating our paper and providing constructive feedback. We acknowledge that the explanation of our method's advantage in node classification was insufficient, and we believe that we have addressed this issue in our response. If any aspects remain unclear or if there are additional questions, we would be happy to address them.
>
> >**W1: The contribution of this paper is somewhat incremental - the idea of using neighborhood information to perform label smoothing has already been extensively studied.**
>
> >**W3: While the proposed method shows superior emprical performance, this paper fail to provide an in-depth explaination on why label-smoothing performs so well in graph-structured data - does it stems from the same reason as i.i.d. data? What makes the proposed method superior than other label smoothing methods on graph?**
>
> We provide an in-depth analysis of the underlying assumptions of PosteL and a case study comparing smoothing methods and PosteL in the **general response at the top of the page**. Please refer to the general response for details.
>
> Firstly, PosteL is significantly distinct from other smoothing methods. We shows that PosteL assigns a high probability to the ground truth label for both homophilic and heterophilic graphs. This is in evident difference to SALS, which is dominated by negative labels on heterophilic graphs.
>
> Comparing PosteL to ALS is non-trivial due to the trainable component in ALS. However, we found that ALS fails to distinguish nodes with different labels when individual node homophily is zero. We also show that PosteL can distinguish the cases.
>
> Based on these differences, we argue that PosteL is significantly distinguished from other smoothing methods. This difference comes from the use of global statistics, $\hat{P}(Y_j=m|Y_i=n, (i,j) \in \mathcal{E})$, so we conjecture that it is the main factor behind PosteL's superior performance. This aligns with the results in Table 2 of our paper, which show that replacing global statistics with local statistics decreases performance.
>
> Next, PosteL assumes that nodes with similar neighborhood label distributions should exhibit similar characteristics. As shown in the general response, this assumption holds for both homophilic and heterophilic graphs. We conjecture that this generality of the assumption contributes to PosteL's strong performance across both types of datasets.
>
> >**W2: "Under the assumption that the neighborhood labels are conditionally independent given the label of the node to be relabeled...", I fail to comprehend this statement, hope authors can further explain this. But in any case, I think the assumption of two adjacent nodes are conditioally independent is too strong.**
>
> The statement is used to derive the factorization of the joint distribution in lines 153-156 as follows $P(\\{Y_j = y_j|j\in\mathcal{N}(i)\\}|Y_i=k) \approx \prod_{j \in \mathcal{N}(i)} {P}(Y_j=y_j | Y_i = k, (i,j)\in\mathcal{E})$.
> We have clarified this assumption in revised version (line 151-153). This assumption might be stong, however this kind of assumption is widely adopted in machine learning in general to make the model simple and tractable. One example is the i.i.d. assumption between data points, which are widely accepted even though it is unclear whether it holds in many cases. In the context of graph learning, [1,2,3] assume that L-hop ego graph of a target node is i.i.d., which is similar to our assumption.
>
> >**W4: The improvements over the most competitive baselines are marginal - for most cases the differences are within the standard deviation.**
>
> Compared to ALS, which achieves the second-best performance, PosteL outperforms it beyond the 95% confidence interval in 13 out of 70 settings. Therefore, we argue that PosteL is significantly better than even the most competitive baselines.
>
> [1] Verma, Saurabh, and Zhi-Li Zhang. "Stability and generalization of graph convolutional neural networks." KDD 2019.
> [2] Garg, Vikas, Stefanie Jegelka, and Tommi Jaakkola. "Generalization and representational limits of graph neural networks." ICML 2020.
> [3] Wu, Qitian, et al. "Handling distribution shifts on graphs: An invariance perspective." arXiv preprint 2022.

---

> ### Author Response · Authors · 2024-11-29
>
> Whenever you have a chance, we’d appreciate your feedback on our rebuttal - thank you.

---

### Author Response · Authors · 2024-11-21
**General response**

Thanks to all the reviewers for their dedicated efforts during the rebuttal phase. We have carefully considered all of the comments, and we have revised our manuscript to incorporate the reviewers' feedback.

### **In-depth analysis of the underlying assumptions of PosteL**
In the first response, we aim to provide an in-depth explanation of the underlying assumption of PosteL. We have included this analysis in Appendix A of the revised version.

Our posterior distribution, $P(Y_i=k|\\{Y_j = y_j|j\in\mathcal{N}(i)\\})$, is based on the assumption that nodes with similar neighborhood label distributions should exhibit similar characteristics. Recently, [1] introduced neighborhood context similarity, $\mathcal{S}(\mathcal{G})$, defined as:

\begin{equation}
\mathcal{S}(\mathcal{G})=\sum_{k=1}^{K}\sum_{u,v\in\mathcal{V_k}}\cos(d(u),d(v)),
\end{equation}

where $K$ is the number of classes, $\mathcal{V}_k$ is the set of nodes with label $k$, $d(u)$ is the neighborhood label distribution, and $\cos(\cdot)$ is the cosine similarity. (We omit some terms for simplicity.) $\mathcal{S}(\mathcal{G})$ represents the degree to which the neighborhood distributions between nodes with the same label are similar. This metric is closely related to our assumption: if $\mathcal{S}(\mathcal{G})$ is large, our assumption holds. The table below, from [1], presents the edge homophily ratio $\mathcal{H}(\mathcal{G})$ and neighborhood context similarity $\mathcal{S}(\mathcal{G})$ for each dataset.

||Cora|CiteSeer|Pubmed|Computers|Photo|Chameleon|Actor|Squirrel|Texas|Cornell|
|-|-|-|-|-|-|-|-|-|-|-|
|$\mathcal{H}(\mathcal{G})$|0.81|0.74|0.80|0.78|0.83|0.23|0.22|0.22|0.11|0.30|
|$\mathcal{S}(\mathcal{G})$|0.89|0.81|0.87|0.90|0.91|0.67|0.68|0.73|0.79|0.40|

This table shows that our assumption generally holds across both homophilic and heterophilic graphs. We conjecture that this generality of the assumption contributes to PosteL's strong performance on both homophilic and heterophilic datasets.

### **The comparison between PosteL and the other smoothing methods**

In the second response, we provide differences between PosteL and other label smoothing methods from a different perspective, especially on heterophilic graphs. We have included this analysis in Appendix C of the revised version.

**1. The effect of utilizing global statistics**
PosteL leverages global statistics, specifically $\hat{P}(Y_j=m|Y_i=n, (i,j)\in\mathcal{E})$ and $\hat{P}(Y_i=m)$, to generate soft labels. In contrast, SALS [2] only utilizes information from 1-hop neighbors. The use of global statistics in ALS [3] is questionable due to the presence of learnable component in their soft label. We provide an example of the soft labels on a binary node classification task with a bipartite graph at this [**(anonymized link)**](https://ibb.co/gVD9rBs). The toy example highlights the key differences between existing methods and ours, which will be elaborated further below.

**1-1. Conditional probability $\hat{P}(Y_j=m|Y_i=n)$ and its impact**
$\hat{P}(Y_j=m|Y_i=n, (i,j)\in\mathcal{E})$ is the conditional probability of a label given the neighborhood label. We analyze the effect of the conditional distribution in balanced binary classification. Let us define node-wise individual homophily $p_i$ and class homophily $c_k$ as:

\begin{equation}
p_i:=\frac{|\\{(i,j)|(i,j)\in\mathcal{E},y_i=y_j\\}|}{|\\{(i,j)|(i,j)\in\mathcal{E}\\}|},
\end{equation}

\begin{equation}
c_k:=\frac{|\\{(i,j)|(i,j)\in\mathcal{E},y_i=k,y_j=k\\}|}{|\\{(i,j)|(i,j)\in\mathcal{E},y_i=k\\}|}=\hat{P}(Y_j=k|Y_i=k, (i,j)\in\mathcal{E}).
\end{equation}

With PosteL, the probability that the posterior label is the same as the ground truth label is given by:

\begin{equation}
P(Y_i=y_i|\\{Y_j = y_j|j\in\mathcal{N}(i)\\})  = \frac{c_{y_i}^{|\mathcal{N}(i)|p_i}(1-c_{y_i})^{|\mathcal{N}(i)|(1-p_i)}}{c_{y_i}^{|\mathcal{N}(i)|p_i}(1-c_{y_i})^{|\mathcal{N}(i)|(1-p_i)}+c_{y_i'}^{|\mathcal{N}(i)|(1-p_i)}(1-c_{y_i'})^{|\mathcal{N}(i)|p_i}},
\end{equation}

where $y_i$ is the ground truth label of node $i$ and $y_i'$ is the other label.

With homophilic graphs where $c_y>(1-c_y), c_{y'}>(1-c_{y'})$ and $p_i > (1-p_i)$ generally, the posterior distribution of the ground truth label is higher than the negative label. With heterophilic graphs, where $c_y<(1-c_y), c_{y'}<(1-c_{y'})$ and $p_i < (1-p_i)$ generally, the posterior distribution of the ground truth is also higher than the negative label.

A simple analysis shows that PosteL assigns a high probability to the ground truth label regardless of whether the graph is homophilic or heterophilic. The presence of the global label statistics $\hat{P}(Y_j=m|Y_i=n, (i,j)\in\mathcal{E})$ plays an important role in these cases, as they provide information about the overall homophilicity of the graph.

---

> ### Author Response · Authors · 2024-11-21
> **General response**
>
> **SALS [2]**
> SALS interpolates the ground truth label with the neigbhorhood labels to generate a soft label. Hence, when the graph is heterophilic, the soft label is likely to be dominated by the negative labels. The figure in the [anonymized link](https://ibb.co/gVD9rBs) shows the example of SALS, where the soft label is dominated by the negative label.
>
> **ALS [3]**
> The label smoothing of ALS consist of three processes: 1) ALS aggregates the neighborhood labels using the formula $\frac{1}{|\mathcal{N}(i)|}\sum_{j\in\mathcal{N}(i)}\mathbf{e}_j$, 2) the aggregated labels are interpolated with the ground truth label, and 3) the interpolated labels are transformed via a linear transform parameterized by learnable weight matrix $\mathbf{W}$ followed by softmax. (Note that our analysis is based on the published version of ALS [3] and not arxiv version - the methods are slightly different between these two versions.)
>
> Analyzing the deterministic behavior of ALS [3], as done previously for PosteL and SALS, is non-trivial due to the presence of a learnable component. However, there are explicit cases where ALS fails to distinguish nodes with different labels. When $p_i=0$, ALS generates a soft label as $\text{softmax}([1,1]\mathbf{W})$, regardless of the value of $y_i$. In such cases, ALS adds the same noise regardless of the characteristics of nodes, which is identical to uniform smoothing. The [attached figure](https://ibb.co/gVD9rBs) illustrates a example of ALS, where it assigns the same label to all nodes.
>
> **Discussion**
> Based on these differences, we argue that PosteL is significantly distinguished from other smoothing methods. This difference comes from the use of global statistics, $\hat{P}(Y_j=m|Y_i=n, (i,j) \in \mathcal{E})$, so we conjecture that it is the main factor behind PosteL's superior performance. This aligns with the results in Table 2 of our paper, which show that replacing global statistics with local statistics decreases performance.
>
> **1-2. Global label distribution $\hat{P}(Y_i=m)$ and imbalanced datasets**
> Next, $\hat{P}(Y_i=m) := \frac{\lvert \set{ u \mid y_u = m } \rvert}{\lvert \mathcal{V} \rvert}$, represents the proportion of label $m$ across all nodes. This term is multiplied by the conditional probability, so when a label has a low proportion, PosteL assigns a lower probability to it. Reflecting the label distribution across the entire graph can be advantageous for graphs with imbalanced labels. For example, PosteL demonstrates consistent performance improvement on the imbalanced Computers dataset, where the label proportions are shown in the following table.
>
> ||1|2|3|4|5|6|7|8|9|10|
> |-|-|-|-|-|-|-|-|-|-|-|
> |**Proportion(%)**|3.2|15.6|10.3|4.0|37.5|2.2|3.5|6.0|15.7|2.1|
>
> **1-3. Further analysis on global statistics**
> Additionally, We demonstrate in subsection **[Influence of neighborhood label distribution]** in Section 4.2 that global label statistics contain sufficient information for distinguishing labels. Furthermore, we show in Table 2 that relying solely on local information negatively impacts performance. Therefore, we argue that the use of global statistics is the reason for PosteL's superior performance.
>
> **2. The advantage of utilizing pseudo labeling strategy**
> Another distinction is the pseudo labeling strategy. SALS and ALS cannot work on sparse graphs, as there will be no labeled neighborhoods. Our pseudo labeling strategy enables smoothing even on sparse graphs by assigning pseudo label to unlabeled nodes. To the best of our knowledge, PosteL is the first label smoothing approach in node classification to address sparse label scenarios.
>
> [1] Xiao, Teng, et al. "Simple and asymmetric graph contrastive learning without augmentations." NeurIPS 2024.
> [2] Wang, Yiwei, et al. "Structure-aware label smoothing for graph neural networks." arXiv preprint 2021.
> [3] Zhou, Kaixiong, et al. "Adaptive label smoothing to regularize large-scale graph training." Proceedings of the 2023 SIAM International Conference on Data Mining (SDM). Society for Industrial and Applied Mathematics, 2023.

---

> ### Author Response · Authors · 2024-11-28
> **General response**
>
> ### **Theoretical analysis of the characteristics of PosteL with heterophilic graphs**
>
> We provide additional two lemmas showing the characteristics of PosteL with heterophilic graphs.
>
> For simplicity, we focus on a binary classification problem with two classes: 0 and 1. The posterior label of the node being labeled to 0 is:
>
> \begin{equation}
> P(Y_i=0|\\{Y_j = y_j|j\in\mathcal{N}(i)\\})  = \frac{c_0^{|\mathcal{N}(i)|p_i}(1-c_0)^{|\mathcal{N}(i)|(1-p_i)}}{c_0^{|\mathcal{N}(i)|p_i}(1-c_0)^{|\mathcal{N}(i)|(1-p_i)}+c_1^{|\mathcal{N}(i)|(1-p_i)}(1-c_1)^{|\mathcal{N}(i)|p_i}}.
> \end{equation}
>
> **Lemma 1. In a heterophilic graph with $c_0, c_1 < 0.5$, if two nodes have the same degree $d$, and node $i$ is connected to more nodes labeled 1 than node $j$, then PosteL assigns a higher probability of node $i$ being labeled 0 compared to node $j$.**
>
> Since the number of adjacent nodes with a different label is larger for node $i$, we have $p_i<p_j$. The lemma can be expressed as follows:
>
> \begin{equation}
> \frac{c_0^{dp_i}(1-c_0)^{d(1-p_i)}}{c_0^{dp_i}(1-c_0)^{d(1-p_i)}+c_1^{d(1-p_i)}(1-c_1)^{dp_i}}>\frac{c_0^{dp_j}(1-c_0)^{d(1-p_j)}}{c_0^{dp_j}(1-c_0)^{d(1-p_j)}+c_1^{d(1-p_j)}(1-c_1)^{dp_j}}.
> \end{equation}
>
> Expanding the inequality:
>
> \begin{align}
> c_0^{d(p_i+p_j)}(1-c_0)^{d(2-p_i-p_j)}+c_0^{dp_i}(1-c_0)^{d(1-p_i)}c_1^{d(1-p_j)}(1-c_1)^{dp_j} >\\
> c_0^{d(p_i+p_j)}(1-c_0)^{d(2-p_i-p_j)}+c_0^{dp_j}(1-c_0)^{d(1-p_j)}c_1^{d(1-p_i)}(1-c_1)^{dp_i}.
> \end{align}
>
> Subtracting $c_0^{d(p_i + p_j)}(1 - c_0)^{d(2 - p_i - p_j)}$ from both sides:
> \begin{equation}
>     c_0^{dp_i}(1-c_0)^{d(1-p_i)}c_1^{d(1-p_j)}(1-c_1)^{dp_j}>c_0^{dp_j}(1-c_0)^{d(1-p_j)}c_1^{d(1-p_i)}(1-c_1)^{dp_i}.
> \end{equation}
>
> \begin{equation}
>     ((1-c_0)(1-c_1))^{d(p_j-p_i)}>(c_0c_1)^{d(p_j-p_i)}.
> \end{equation}
>
> Since $c_0<1-c_0$ and $c_1<1-c_1$ imply that $(1 - c_0)(1 - c_1)>c_0c_1$, and given $(1 - c_0)(1 - c_1)>c_0c_1$ and $p_j-p_i> 0$, the inequality holds.
>
> **Lemma 2. In a heterophilic graph with $c_0, c_1 < 0.5$, if two nodes are connected only to nodes labeled 1, and their respective degrees are $n$ and $m$ ($n > m$), then PosteL assigns a higher probability of being labeled 0 to the node with the higher degree.**
>
> When nodes are connected only to nodes labeled 1, PosteL assigns the posterior probability of the node being labeled 0 as follows:
> \begin{equation}
>     P(Y_i=0|\\{Y_j = 1|j\in\mathcal{N}(i)\\})=  \frac{(1-c_0)^{\text{deg}(i)}}{(1-c_0)^{\text{deg}(i)}+c_1^{\text{deg}(i)}},
> \end{equation}
>
> where $\text{deg}(i)$ represents the degree of node $i$. The lemma can be expressed as follows:
> \begin{equation}
>     \frac{(1-c_0)^n}{(1-c_0)^n+c_1^n} > \frac{(1-c_0)^m}{(1-c_0)^m+c_1^m}.
> \end{equation}
>
> Expanding the inequality:
>
> \begin{equation}
>     ((1-c_0)^m+c_1^m)(1-c_0)^n > ((1-c_0)^n+c_1^n)(1-c_0)^m.
> \end{equation}
>
> \begin{equation}
>     (1-c_0)^n(1-c_0)^m+c_1^m(1-c_0)^n > (1-c_0)^n(1-c_0)^m+c_1^n(1-c_0)^m.
> \end{equation}
>
> \begin{equation}
>     c_1^m(1-c_0)^n > c_1^n(1-c_0)^m.
> \end{equation}
>
> \begin{equation}
>     (1-c_0)^{n-m} > c_1^{n-m}.
> \end{equation}
>
> Since $c_0,c_1<0.5$, it follows that $1-c_0>c_1$. Thus, the inequality holds.
>
> As a corollary we can show the following two properties.
>
> 1. If two nodes have the same degree, the node connected to more nodes labeled 1 should have a higher probability of being labeled 0.
> 2. If two nodes are connected only to nodes labeled 1, the node with the higher degree should have a higher probability of being labeled 0.
>
> Although these lemmas and corollary may not reflect the real-world scenario, analyzing properties of a model is an important step towards understanding its performance.

---

### Meta-Review · Area_Chair_6Po9 · 2024-12-19

**Metareview:**

This paper proposes a label smoothing method for transductive node classification, which incorporates local context through neighborhood label distribution. However, the reviewers provided negative feedback, raising several concerns, including: (1) the assumption that neighborhood labels are conditionally independent given the node's label is unclear; (2) the improvements over competitive baselines are viewed as marginal; and (3) the method faces limitations in handling heterogeneous graphs with sparse labels. Based on these considerations, the paper is not recommended for acceptance at this time.

**Additional Comments On Reviewer Discussion:**

During the rebuttal period, the reviewers' opinions remained unchanged.

---

### Decision · Program_Chairs · 2025-01-22

Reject